Resource

# GRAPE for fast and scalable graph processing and random-walk-based embedding

Luca Cappelletti [1], Tommaso Fontana [1], Elena Casiraghi[1,2,3], Vida Ravanmehr[4,5], Tiffany J. Callahan [6], Carlos Cano[7], Marcin P. Joachimiak[3], Christopher J. Mungall[3], Peter N. Robinson [4], Justin Reese[3] & Giorgio Valentini [1,2,8,9] ✉

Graph representation learning methods opened new avenues for addressing complex, real-world problems represented by graphs. However, many graphs used in these applications comprise millions of nodes and billions of edges and are beyond the capabilities of current methods and software implementations. We present GRAPE (Graph Representation Learning, Prediction and Evaluation), a software resource for graph processing and embedding that is able to scale with big graphs by using specialized and smart data structures, algorithms, and a fast parallel implementation of random-walk-based methods. Compared with state-of-the-art software resources, GRAPE shows an improvement of orders of magnitude in empirical space and time complexity, as well as competitive edge- and node-label prediction performance. GRAPE comprises approximately 1.7 million well-documented lines of Python and Rust code and provides 69 node-embedding methods, 25 inference models, a collection of efficient graph-processing utilities, and over 80,000 graphs from the literature and other sources. Standardized interfaces allow a seamless integration of third-party libraries, while ready-to-use and modular pipelines permit an easy-to-use evaluation of graph-representation-learning methods, therefore also positioning GRAPE as a software resource that performs a fair comparison between methods and libraries for graph processing and embedding.

In various fields such as biology, medicine, and data and network science, graphs can naturally model available knowledge as interrelated concepts, represented by a network of nodes connected by edges. The wide range of graph applications has motivated the development of a rich literature on graph representation learning (GRL) and inference models[1].

GRL models compute embeddings, that is, vector representations of the graph and its constituent elements, capturing their topological, structural, and semantic relationships. Graph inference models can use such embeddings and available additional features for several tasks, for example, visualization, clustering, and prediction of node labels, edges and edge labels[1]. State-of-the-art GRL algorithms, including, among

[1]AnacletoLab, Dipartimento di Informatica, Università degli Studi di Milano, Milan, Italy. [2]National Laboratory in Artificial Intelligence and Intelligent Systems, Consorzio Interuniversitario Nazionale per l'Informatica, Rome, Italy. [3]Division of Environmental Genomics and Systems Biology, Lawrence Berkeley National Laboratory, Berkeley, CA, USA. [4]The Jackson Laboratory for Genomic Medicine, Farmington, CT, USA. [5]Department of Lymphoma and Myeloma, The University of Texas MD Anderson Cancer Center, Houston, TX, USA. [6]Department of Biomedical Informatics, Columbia University Irving Medical Center, New York, NY, USA. [7]Department of Computer Science and Artificial Intelligence, University of Granada, Granada, Spain. [8]European Laboratory for Learning and Intelligent Systems, Tübingen, Germany. [9]Data Science Research Center, Università degli Studi di Milano, Milan, Italy. ✉e-mail: valentini@di.unimi.it

others, methods based on matrix factorization, random walks (RWs), graph kernels[2], triple sampling, and (deep) graph neural networks (GNNs)[1,3], have shown their effectiveness in analyzing networks from sociology, biology, medicine, and many other disciplines. Although a great deal of research has been devoted to the development of software resources for graph processing and analysis (for example, iGraph[4], GraphLab[5], NetworkX[6], GraphX[7], and SNAP[8]) or for GRL (for example, PecanPy[9], PyKeen[10], DGL[11], Pytorch Geometric[12], and Spektral[13]), real-world networks often include millions of nodes and billions of edges, thus raising the problem of the scalability of existing software resources[14]. In particular, the scalability of GNNs represents an open issue[3], despite recent efforts to design GNNs that can scale with large graphs[15].

In this context, for scalability issues, RW-based GRL models are often preferred. However, their performance is often affected by the high computational costs required by the RW generators. Indeed, current state-of-the-art RW-based graph-embedding libraries display a limited ability to efficiently generate enough RW data samples to accurately represent the topology of the underlying graph. This limits the performance of node- and edge-label prediction methods, which strongly depends on the informativeness of the underlying embedded graph representation. The efficient generation of billions of sampled RWs could lead to more accurate embedded representations of graphs and could boost the performance of machine learning methods that learn from the embedded vector representation of nodes and edges.

The findable, accessible, interoperable, and reusable (FAIR) comparison of different graph-based methods under different experimental set-ups is a relevant open issue, only very recently considered in literature in the context of the Open Graph Benchmark Large-Scale Challenge (OGB-LSC). This initiative enables a FAIR comparative evaluation of different models on three specific large-scale graphs[16]. However, further efforts are required to provide standard interfaces to easily integrate methods from different libraries and public experimental pipelines, and to allow a FAIR comparison of different methods and libraries for the analysis of any graph-based data.

GRAPE provides a modular and flexible solution to the above problems by offering (1) a scalable and fast software library that efficiently implements RW-based embedding methods, graph-processing algorithms, and inference models that can run on both general-purpose desktop and laptop computers, as well as on high-performance computing clusters; (2) an extensive set of efficient and effective built-in GRL algorithms that any user can continuously update by implementing easy-to-use standardized interfaces; and (3) ready-to-use evaluation pipelines to provide a fair and reproducible evaluation of any GRL algorithm (implemented or integrated into GRAPE) using the ~80,000 graphs retrievable through the library and also other graphs provided by the user. Therefore, GRAPE can also be viewed as an efficient collector of GRL methods that can perform a FAIR comparison on a large set of available graphs.

## Results
### Embiggen and Ensmallen
GRAPE consists of approximately 1.5 million lines of Python code and approximately 200,000 lines of Rust code (results computed with the Tokei tool, https://docs.rs/tokei/latest/tokei), implementing efficient data structures and parallel computing techniques to enable scalable graph processing and embedding.

The library's high-level structure, overall functionalities, and two core modules, Ensmallen (Enabler of Small Computational Resources for Large Networks) and Embiggen (Embedding Generator), are depicted in Fig. 1a.

Ensmallen efficiently loads big graphs and executes graph-processing operations, owing to its Rust[17] implementation and to the usage of map-reduce thread-based parallelism and branch-less single-instruction multiple data (SIMD) parallelism. It also provides Python bindings for ease of use.

Designed to leverage succinct data structures[18], GRAPE requires only a fraction of the memory required by other libraries and guarantees average constant-time rank-and select operations[19]. This makes it possible to execute many graph-processing tasks, for example, accessing node neighbors and running first- and second-order RWs, with memory usage close to the theoretical minimum.

However, the performance of RW-based embedding methods is often affected by the high computational costs required by the RW generators that often rely on a limited number of RW samples that cannot accurately represent the topology of the underlying graph. This leads to uninformative graph embeddings that affect the performance of the subsequent graph-prediction models. To overcome these limitations, GRAPE focuses on smart and efficient implementations of RW-based embedding methods since its main objective is to scale with large graphs (see Methods for details), while other effective but more complex models based, for example, on GNNs[3] available from other libraries[20] are not yet implemented in the library because of their well-known scaling limitations[3,21].

Among the many high-performance algorithms implemented in GRAPE, we propose an algorithm, sorted unique sub-sampling (SUSS), that allows approximated RWs to be computed to enable the processing of graphs that contain very-high-degree nodes (degree $> 10^6$), unmanageable for the corresponding exact analogous algorithms. Approximated RWs can achieve edge prediction performance comparable to those obtained by the corresponding exact algorithm with a speed-up from two to three orders of magnitude.

Ensmallen also provides many other methods and utilities, such as refined multiple holdout techniques to avoid biased performance evaluations; Bader and Kruskal algorithms for computing random and minimum spanning arborescence and connected components; stress and betweenness centrality[22]; node and edge filtering methods; and algebraic set operations on graphs. Ensmallen allows graphs to be loaded from a wide variety of node and edge list formats. In addition, users can automatically load data from an ever-increasing list of over 80,000 graphs from the literature and elsewhere (Fig. 1b).

Embiggen provides efficient implementations of GRL and inference models, including an exhaustive set of node-embedding methods, for example, spectral and matrix factorization models such as High-Order Proximity preserved Embedding (HOPE)[23], Network Embedding as Matrix Factorization (NetMF)[24], and their variations (Geometric Laplacian Eigenmap Embedding - GLEE)[25], SocioDim[26]). Moreover, it offers from-scratch implementations of Continuous Bag of Words (CBOW), SkipGram, and GloVe embedding methods[27,28], which substantially outperform the Keras-based ones, as Tensor-Flow Application Programming Interfaces (APIs) are too coarse and high level for such fine-grained optimizations. GRAPE implements RW-based methods such as DeepWalk, Node2Vec, and Walklets[29,30]; triple-sampling methods such as Large-scale Information Network Embedding (LINE)[31] and corrupted-triple-sampling methods such as Translating Embeddings (TransE)[32]; and, more generally, a wide range of inference methods.

GRAPE provides three modular pipelines to compare and evaluate node-label, edge-label, and edge prediction performance under different experimental settings (Fig. 1b), as well as utilities for graph visualization (Fig. 1c). These pipelines allow non-expert users to tailor their desired experimental set-up and quickly obtain actionable and reproducible results (Fig. 1b). Furthermore, GRAPE provides interfaces to integrate third-party models and libraries (for example, Karate Club[33] and PyKeen[10] libraries). This way, the evaluation pipelines can compare models implemented or integrated into GRAPE.

The possibility of integrating external models and the availability of graphs for testing them on the same datasets allow the answering of a still open and crucial issue in literature, which is regarding the FAIR, objective, reproducible, and efficient comparison of graph-based methods and software implementations.

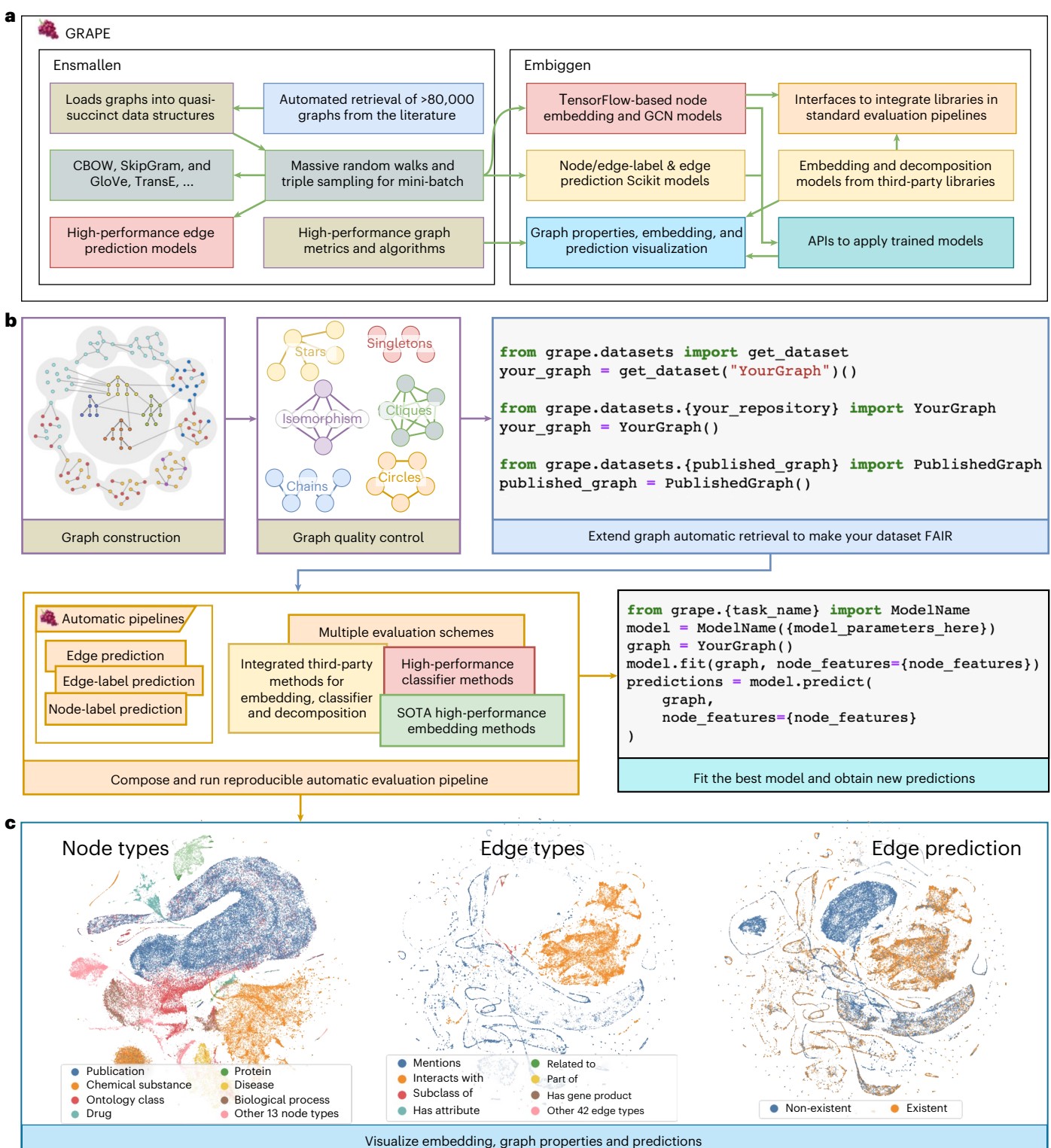

**Fig. 1 | Schematic of GRAPE, Ensmallen and Embiggen. a**, High-level structure of the GRAPE software resource. **b**, Pipelines for an easy, fair, and reproducible comparison of graph-embedding techniques, graph-processing methods, and libraries. **c**, Visualization of the KGCOVID19 graph[53], obtained by displaying the first two components of the t-distributed stochastic neighbor embedding computed by using a Node2Vec SkipGram model that ignores the node and edge type during the computation. The clusters' colors indicate the Biolink category for each node (left), the Biolink category for each edge (center), and the predicted edge existence (right).

We used the evaluation pipelines to compare the edge- and node-label prediction performance of 16 embedding models. Moreover, we compared GRAPE with state-of-the-art graph-processing libraries across several types of graphs having different sizes and characteristics, including big real-world graphs such as Wikipedia, the Comparative Taxonomic Database (CTD)[34], and biomedical knowledge

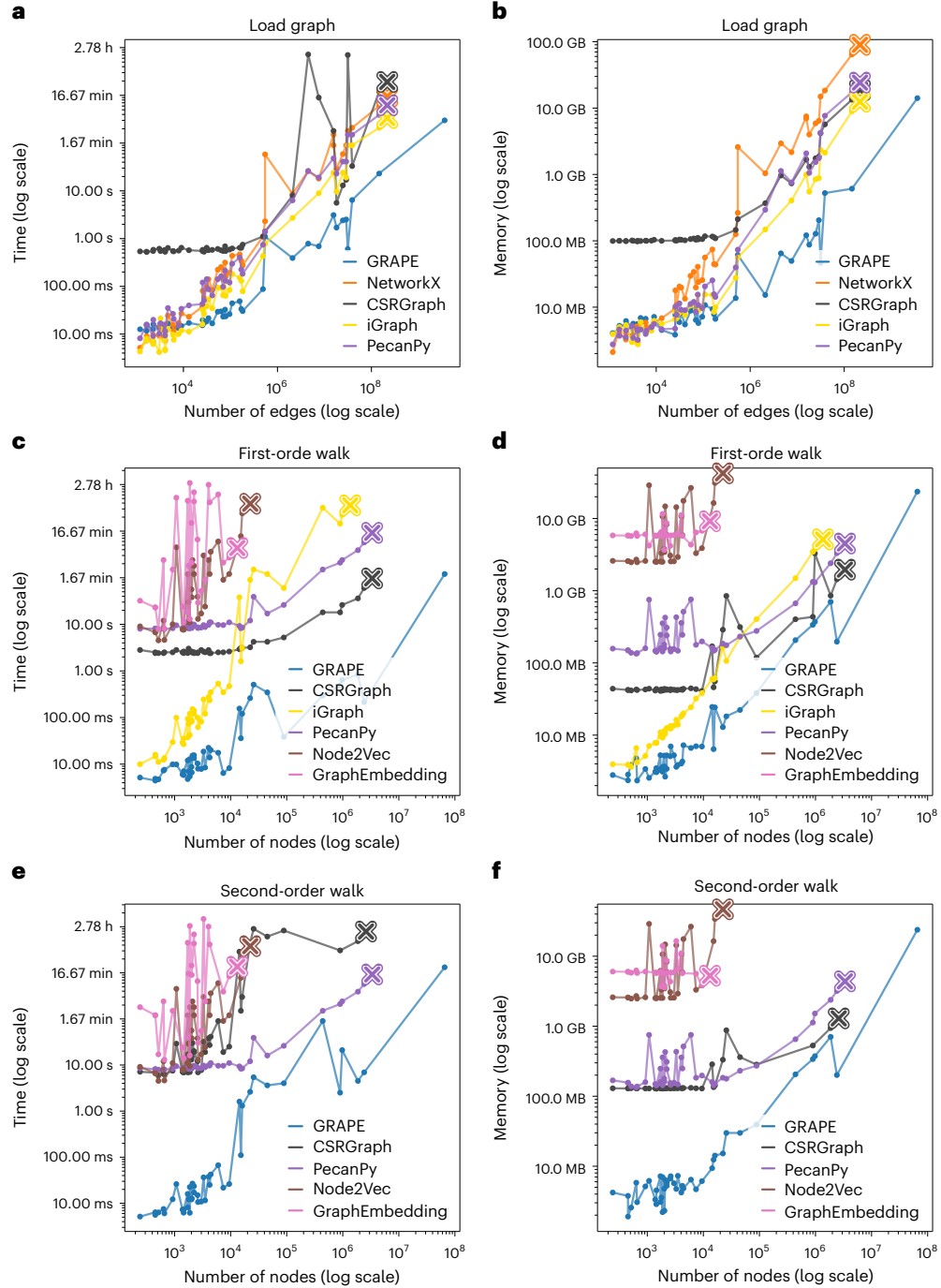

**Fig. 2 | Experimental comparison of GRAPE with state-of-the-art graph-processing libraries across 44 graphs. a,b**, Graph loading. **a**, Empirical execution time. **b**, Peak memory usage; the horizontal axis shows the number of edges, and the vertical axis shows peak memory usage. **c,d**, First-order RW. **c** Empirical execution time. **d**, Peak memory usage. **e,f**, Second-order RW.

**e**, Empirical execution time. **f**, Peak memory usage. The multiplication symbols represent when a library crashes, exceeds 200 GB of memory, or takes more than 4 h to execute the task. Each line corresponds to a graph resource/library, and points on the lines refer to the 44 graphs used in the experimental comparison.

graphs generated through PheKnowLator[35], showing that GRAPE achieves state-of-the-art results in processing big real-world graphs both in terms of empirical time and space complexity and prediction performance.

## Fast error-resilient graph loading

GRAPE has been carefully designed to efficiently perform in space and time. In this section, we carried out a comparative study of performance with state-of-the-art graph-processing libraries (including

NetworkX[36], iGraph[4], CSRGraph, PecanPy[9]) in terms of empirical space and time used for loading 44 different real-world graphs (Fig. 2a,b). Results show that GRAPE is faster and requires less memory than the state-of-the-art libraries. For instance, GRAPE loads the ClueWeb09 graph (1.7 billion nodes and 8 billion undirected edges) in less than 10 min and requires approximately 60 GB of memory, whereas the other libraries were not able to load this graph. In addition, GRAPE can process many graph formats and check for common format errors simultaneously. All graphs and libraries used in these experiments are

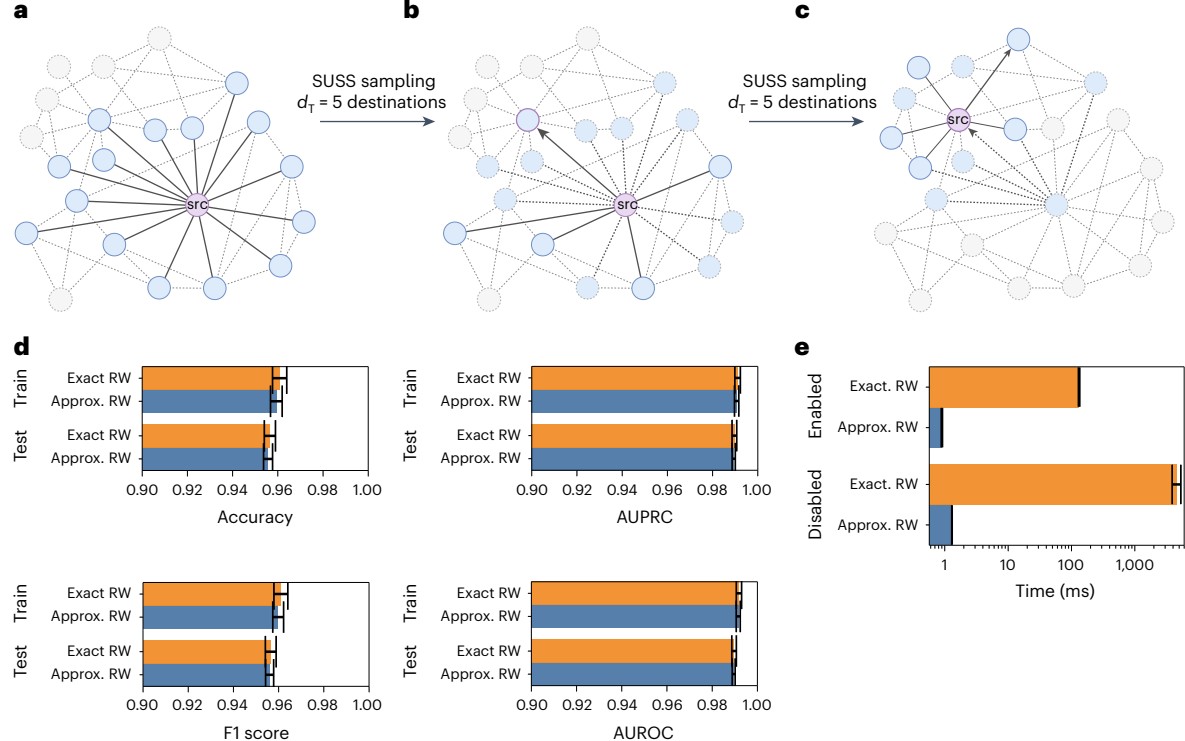

**Fig. 3 | Approximated RW. a,** The RW starts at the source node src; its 15 neighborhood nodes are highlighted in cyan. **b,** We sampled $d_T$ = 5 destination nodes ($d_T$, degree threshold) from the available 15 destinations, using our SUSS algorithm, and performed a random step (edge highlighted with an arrow). **c,** A further step was then performed on the successor node (that then became the novel source node src), and the same process was repeated until the end of the walk. **d,** Edge prediction performance comparison (accuracy, AUPRC, F1 score, and AUROC computed over $n$ = 10 holdouts—data are presented as mean values ± s.d.) using SkipGram-based embeddings and RW samples obtained with exact and approximated RWs for both the training and the test sets with the STRING–PPI dataset. Bar plots are zoomed in at 0.9 to 1.0, with error bars representing the s.d., computed over 30 holdouts. **e,** Empirical time comparison of the approximated and exact second-order RW algorithm on the graph sk-2005 (ref. 54): 100-step RWs are run on 100 randomly selected nodes. Error bars represent the s.d. across $n$ = 10 repetitions. Data are presented as mean values ± s.d.

directly available from GRAPE. Detailed results are available in the Supplementary Sections 1 and 2.

## GRAPE outperforms state-of-the-art libraries on RW generation

Through extensive use of thread and SIMD parallelism and specialized quasi-succinct data structures, GRAPE outperforms state-of-the-art libraries by one to four orders of magnitude in the computation of RWs, both in terms of empirical computational time and space requirements (Fig. 2c–f). The method used to measure execution time and peak memory usage properly is presented in Supplementary Section 6.3.

Further speed up of second-order RW computation is obtained by dispatching one of the eight optimized implementations of Node2Vec sampling[29]. The dispatching is based on the values of the return and in–out parameters and the type of the graph (weighted or unweighted). GRAPE automatically provides the version best suited to the requested task, with minimal code redundancy. The time performance difference between the least and the most computationally expensive implementations is around two orders of magnitude (Supplementary Section 7.2 and Supplementary Tables 50 and 51).

**Experimental comparison of graph-processing libraries.** We compared GRAPE with a set of state-of-the-art libraries, including GraphEmbedding, Node2Vec, CSRGraph, and PecanPy[9], on a large set of first- and second-order RW tasks. The RW procedures in the GraphEmbedding and Node2Vec libraries use the alias method (Supplementary Section 7.2.3). The PecanPy library also employs the alias method for small-graph-use cases (less than 10,000 nodes). Contrastingly, CSRGraph computes

the RWs lazily using Numba[37]. Similarly, PecanPy leverages the Numba lazy generation for graphs having more than 10,000 nodes. All libraries are further detailed in Supplementary Section 1. Figure 2 shows the experimental results of a complete iteration of 100-step RWs on all the nodes across 44 graphs with edges ranging from thousands to several billion. GRAPE greatly outperforms all the compared graph libraries on both first- and second-order RWs in terms of space and time complexity. Note that GRAPE scales well with the biggest graphs considered in the experiments, whereas the other libraries either crash when exceeding 200 GB of memory or take more than 4 h to execute the task (Fig. 2c–f).

**Approximated RWs to process graphs with high-degree nodes.** RWs on graphs containing high-degree nodes are challenging since multiple paths from the same node must be processed. To overcome this computational burden, GRAPE provides an approximated implementation of weighted RWs that undersamples the neighbors to scale with graphs containing nodes with high degree, for example, with millions of neighbors (Fig. 3a–c). To guarantee scalability, the sampling process is performed by an algorithm (SUSS) that we developed as an alternative to the classic and computationally demanding alias algorithm (Supplementary Section 7.2.3). SUSS is a sampling algorithm that divides a discrete range into $k$ uniformly spaced buckets and randomly samples a value from each bucket to achieve an efficient neighborhood sub-sampling for nodes with a degree $d \gg k$. The obtained values are inherently sorted and unique.

We compared exact and approximated RW samples for the Node2Vec-based SkipGram for the edge prediction problem on the (unfiltered) Protein-Protein Interaction (PPI) *Homo sapiens* graph

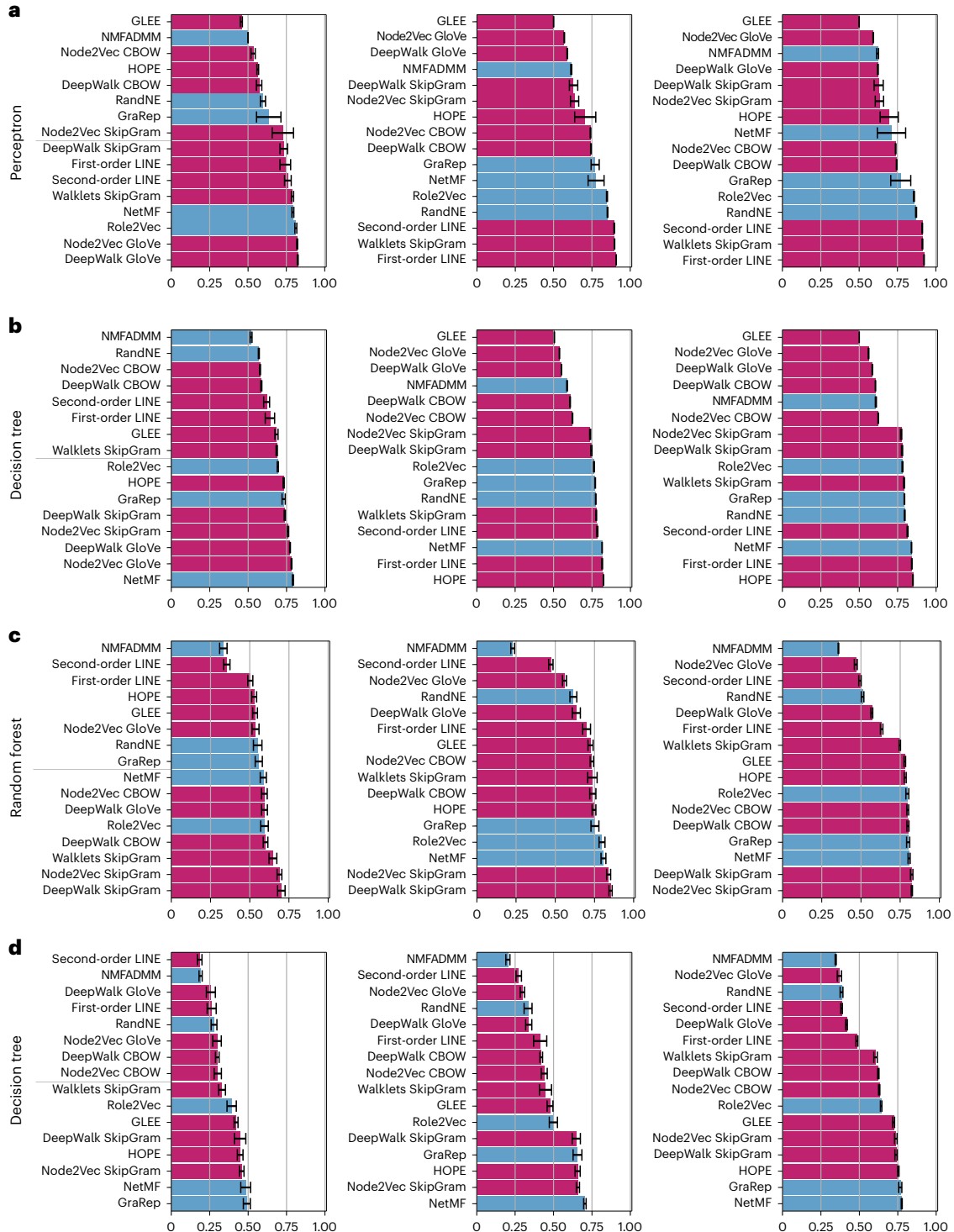

**Fig. 4 | Comparison of embedding methods through the GRAPE pipelines on edge- and node-label prediction.** Results represent the mean balanced accuracy computed across $n = 10$ holdouts ± s.d. (results using other evaluation metrics are available in Supplementary Section 5). We sorted the embedding models by performance for each task; methods directly implemented in GRAPE are in purple, while integrated methods are in cyan. **a**,**b**, Edge prediction results obtained through a perceptron (**a**) and a decision tree (**b**). Bar plots from left to right show the balanced accuracy results obtained with the Human Phenotype Ontology (left), STRING *H. sapiens* (center), and STRING *Mus musculus* (right). **c**,**d**, Node-label prediction results obtained through a random forest (**c**) and a decision tree (**d**). Bar plots from left to right show the balanced accuracy achieved with CiteSeer (left), Cora (center), and PubMed Diabetes (right) datasets.

from the Search Tool for Recurring Instances of Neighbouring Genes (STRING) database[38], achieving a statistically equivalent performance (two-sided Wilcoxon rank–sum $P > 0.2$; Fig. 3d), by running 30 holdouts and setting a (deliberately low) degree threshold equal to 10 for

the approximated RW, while the maximum degree in the training set ranged between 3,325 and 4,184 across the holdouts. These results show no relevant performance decay, even when using a relatively stringent degree threshold.

We used the sk-2005 graph that includes approximately 50 million nodes and 1.8 billion edges and some nodes with degrees over 8 million to better show that approximated RW can be several orders of magnitude faster than the 'vanilla' exact RW algorithm. Indeed, by extrapolating the results reported in Fig. 3e to the entire graph, the exact algorithm requires approximately 23 days, while the approximate one requires approximately 11 min, both running on a PC with two AMD EPYC 7662 64-core processors, 256 CPU threads, and 1 TB RAM.

## GRAPE enables a fair comparison of graph-based methods

GRAPE provides both a large set of ready-to-use graphs that can be used in the experiments and standardized pipelines to fairly compare different models and graph libraries, ensuring reproducibility of the results (Fig. 1b). Graph embedding is efficiently implemented in Rust from scratch (with a Python interface) or is integrated from other libraries by implementing the interface methods of an abstract GRAPE class. GRAPE users can compare different embedding methods and prediction models and add their own methods to the standardized pipelines. Our experiments show how to use the standardized pipelines to fairly compare a large set of methods and different implementations using only a few lines of Python code.

**Experimental comparison of node- and edge-embedding methods.** We selected 16 among the 69 node-embedding methods available in GRAPE, and we used the edge prediction and node-label standardized prediction pipelines to compare the prediction results obtained by the perceptron, decision tree, and random forest classifiers (Fig. 4). We used the Hadamard product for the edge prediction tasks to construct edge embeddings from node embeddings, that is, the element-wise product of the source and destination nodes to obtain the embedding of the corresponding edge. We applied a connected Monte Carlo evaluation schema for edge prediction and a stratified Monte Carlo evaluation schema for node-label prediction (Supplementary Section 10.2).

The models were tested on three graphs for edge prediction (Fig. 4a,b) and three graphs for node-label prediction (Fig. 4c,d). The graph reports, describing the characteristics of the analyzed graphs, automatically generated with GRAPE, are available in Supplementary Sections 3.2 and 3.3. Since they are homogeneous graphs (that is, graphs with only one type of node and edge), we considered only homogeneous node-embedding methods. Moreover, we discarded non-scalable models, for example, models based on the factorization of dense adjacency matrices.

Among the 16 methods, 11 are implemented in GRAPE (purple in Fig. 4) and 5 were integrated from the Karate Club library[33] (cyan in Fig. 4). They can be grouped into four broad classes:

(1) Spectral and matrix factorization methods: geometric Laplacian eigenmap embedding[25], alternating direction method of multipliers for non-negative matrix factorization (NMFADMM)[39], high-order proximity preserved embedding[23], iterative random projection network embedding (RandNE)[40], network matrix factorization[24], and graph representations (GraRep)[41]

(2) First-order RW methods: DeepWalk-based GloVe, CBOW, and SkipGram; Walklets SkipGram[27,28,30]; and Role2Vec with Weisfeiler–Lehman Hashing[2,33,42]

(3) Second-order RW methods: Node2Vec-based GloVe, CBOW, and SkipGram[27–29]

(4) Triple-sampling methods: first- and second-order LINE[31]

Results show that no model is consistently better with respect to the others across the types of tasks and the datasets used in the experiments (Fig. 4). These results are analogous to those obtained[43] for the TransE model family and those obtained[44] for GNN models, highlighting the need for objective pipelines to systematically compare a wide array of possible methods for a desired task. The standardized pipelines implementing the experiments are available from the online GRAPE tutorials and allow the full reproducibility of the results summarized

in Fig. 4. Full results using other evaluation metrics are available in Supplementary Sections 5.1 and 5.2.

## Scaling with big real-world graphs

To show that GRAPE can scale and boost edge prediction in big real-world graphs, we compared its Node2Vec-based models with state-of-the-art implementations on three big graphs: (1) an English Wikipedia graph, (2) a graph constructed using the CTD[34], and (3) a biomedical graph generated through PheKnowLator[35]. Supplementary Section 6.1 reports details about the construction and the characteristics of the three graphs.

**Experimental set-up.** In the experiments, the GRAPE implementations of Node2Vec with both CBOW and SkipGram were compared with those available in the following embedding libraries, widely used by the scientific community: PecanPy[9], NodeVectors (https://github.com/VHRanger/nodevectors), SNAP[8], Node2Vec (https://github.com/eliorc/node2vec), GraphEmbedding (https://github.com/shenweichen/GraphEmbedding), FastNode2Vec (https://github.com/louisabraham/fastnode2vec), and PyTorch Geometric (https://github.com/pyg-team/pytorch_geometric). More details about the above state-of-the-art libraries are reported in Supplementary Section 6.2.

The embeddings computed by each of the tested models were used to train a decision tree available from the Embiggen module of GRAPE for edge prediction. To perform an unbiased evaluation, the training and tests were performed by ten connected Monte Carlo holdouts (with a 80:20 train to test ratio; Supplementary Section 10.2) and performances were evaluated using precision, recall, accuracy, balanced accuracy, F1, Area Under the Receiver Operating Characteristic (AUROC), and Area Under the Precision Recall Curve (AUPRC). In the experimental set-up, we imposed the following memory and time constraints, using a Google Cloud virtual machine with 64 cores and N1 CPUs with an Intel Haswell micro-architecture:

- A maximum time of 48 h for each holdout to produce the embedding
- A 64 GB maximum memory usage allowed during the embedding
- A 256 GB maximum memory usage allowed during the prediction phase

**Results on scaling tests.** GRAPE can scale with big graphs when the other competing libraries fail. Most of the competing libraries could not complete the embedding and prediction tasks on big real-world graphs. Indeed, NodeVectors exceeded the time computation limit, while SNAP, Node2Vec, GraphEmbedding, and PyTorch Geometric went out of memory in the embedding phase, exceeding the available RAM memory (64 GB). By contrast, GRAPE only required 54 MB with the CTD graph. For the first three libraries, this was due to the extremely high memory complexity required by the alias method they use for precomputing the transition probabilities (Supplementary Section 7.2.3); indeed, the alias method has quadratic complexity with respect to the number of nodes in the graph, therefore quickly becoming too expensive on big graphs. We also ran PyTorch Geometric on a substantially smaller graph (the STRING *H. sapiens* graph, having approximately 20,000 nodes and 12 million edges), and we registered that GRAPE is approximately 60 times faster than PyTorch Geometric.

Such a comparison is impossible with the three other libraries employing the alias method, as this smaller graph is still considerably larger than what is possible for them to handle. FastNode2Vec and PecanPy went out of time (more than 48 h of computation) on the biggest Wikipedia graph. In practice, only GRAPE was able to successfully terminate the embedding and prediction tasks with all three big real-world graphs.

GRAPE improves upon the empirical time complexity of state-of-the-art libraries. Figure 5a–c shows the memory and time requirements of

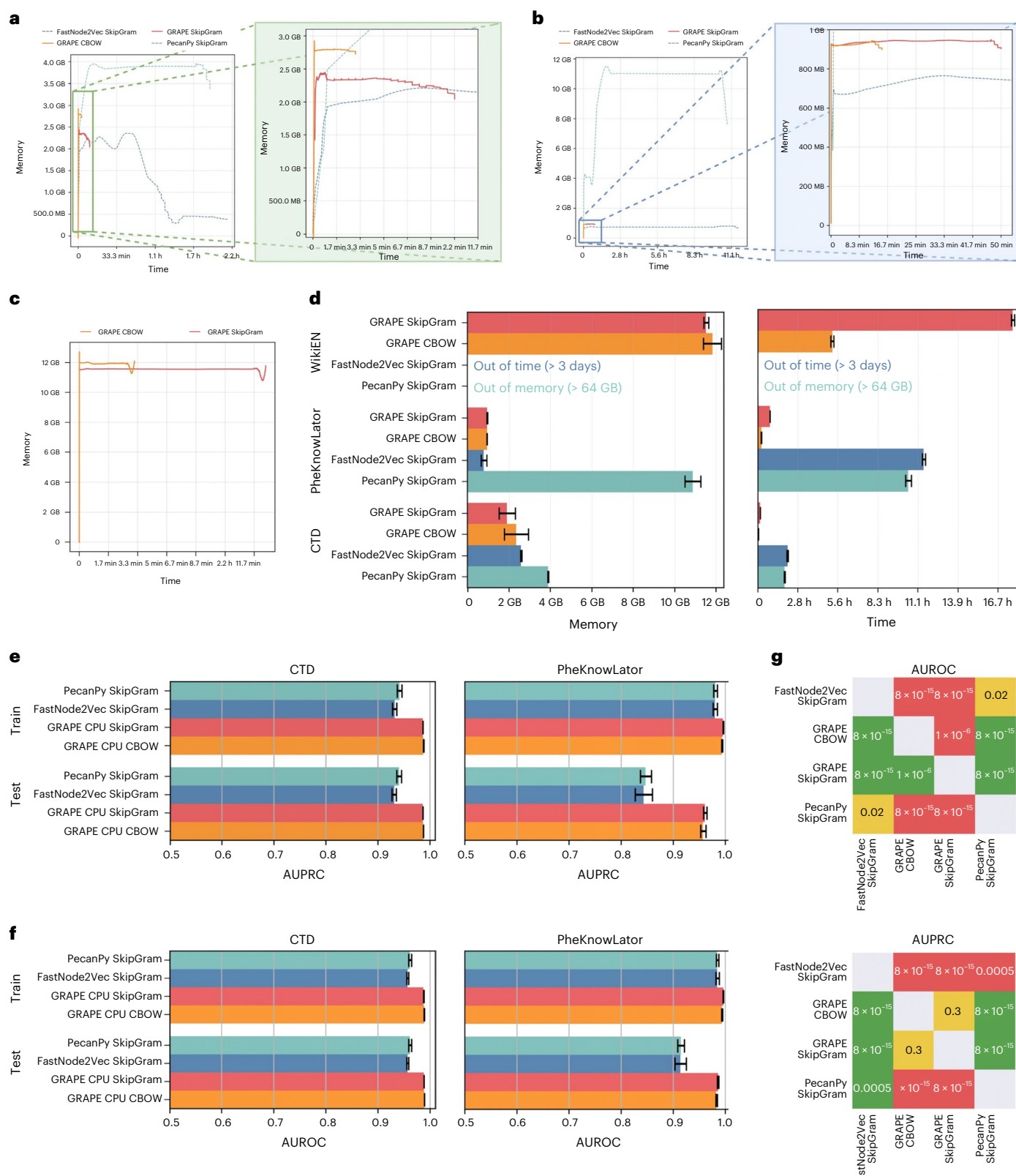

**Fig. 5 | Performance comparison between GRAPE and state-of-the-art implementations of Node2Vec on real-world big graphs.** GRAPE implementations achieve substantially better empirical time complexity. **a–c**, The worst performance (maximum time and memory, denoised using a Savitzky–Golay filter) over ten holdouts on CTD (**a**), PheKnowLator (**b**), and Wikipedia (**c**). In **a** and **b**, the rectangles in the left figure are magnified in the right figure to highlight GRAPE performances. In the Wikipedia plot (**c**), only GRAPE results are available as the others either ran out of time or memory. **d**, Average memory and computational time across $n = 10$ holdouts; data are presented as mean values ± s.d. **e,f**, AUPRC (**e**) and AUROC (**f**) results of decision trees trained with different graph-embedding libraries; data are presented as mean values ± s.d. computed over $n = 10$ holdouts: GRAPE embedding achieves better edge prediction performance than those obtained by the other libraries. **g**, One-sided Wilcoxon signed-rank test results ($P$ values) between GRAPE and the other state-of-the-art libraries, in which a win of a row against a column is in green, a tie in yellow, and a loss in red.

GRAPE, FastNode2Vec, and PecanPy (note that the other state-of-the-art libraries ran out of time or memory on these real-world graph prediction tasks). With the CTD and PheKnowLator biomedical graphs, we can observe a speed-up of approximately one order of magnitude (Fig. 5a,b) of GRAPE with respect to both FastNode2Vec and PecanPy and also a substantial gain in memory usage with respect to PecanPy and a comparable memory footprint with FastNode2Vec. These results are confirmed by the average memory and time requirements across ten holdouts (Fig. 5d). Note that both FastNode2Vec and PecanPy fail with the Wikipedia task, whereas GRAPE was able to terminate the computation in a few hours using a reasonable amount of memory (Fig. 5c,d).

GRAPE boosts edge prediction performance. GRAPE not only allows graph-embedding approaches to be applied to graphs that are bigger than what was previously possible and enables fast and efficient computation, but also can boost prediction performance on big real-world graphs. Figure 5e,f shows that GRAPE achieves better results on edge prediction tasks with both CTD and PheKnowLator biomedical graphs. GRAPE outperforms the other competing libraries at 0.01 significance level, according to the Wilcoxon rank–sum test (Fig. 5g). The edge embeddings have been used to train a decision tree to allow a safe comparison between the embedding libraries.

Supplementary Section 6.4 reports AUROC, accuracy, and F1-score performances and other more detailed results about the experimental comparison of GRAPE with state-of-the-art libraries.

## Discussion

We have presented GRAPE, a software resource with specialized data structures, algorithms, and fast parallel implementations of graph-processing methods coupled with efficient implementations of algorithms for RW-based GRL. Our experiments have shown that GRAPE significantly outperforms state-of-the-art graph-processing libraries in terms of empirical space and time complexity, with an improvement of up to several orders of magnitude for common RW-based analysis tasks. This allows substantially bigger graphs to be analyzed and may improve the performance of graph machine learning methods by allowing for more comprehensive training, as shown by our experiments performed on three real-world large graphs. In addition, the substantial reduction of the computational time achieved by GRAPE in common graph-processing and learning tasks will help to reduce the carbon footprint of machine learning researchers and graph-processing and analyzing practitioners in several disciplines.

Thanks to (1) the huge number of well-known graphs that can be efficiently loaded and used via GRAPE, (2) the standard interfaces that allow any user to integrate their own GRL models into GRAPE, and (3) the modular pipeline that allows the easy design of different benchmarking experiments, GRAPE can be used to perform a FAIR comparison between virtually any methods and using any graph data (including graph data directly provided by the users).

Another related resource that allows a similar comparison is the OGB resource[16]. However, as witnessed by the recent OGB-LSC (https://ogb.stanford.edu/neurips2022/), the datasets and the organization of the OGB resource are well suited for specific large-scale challenges, while the GRAPE evaluation pipelines are useful for assessing and comparing any method on any graph benchmark chosen by any user. This makes the two resources related but complementary in their different purposes.

We would further remark that GRAPE currently provides efficient implementations of RW-based embeddings, whose advantage is their applicability to a larger set of learning problems since the computed embeddings are usually task independent and unsupervised. By contrast, embeddings computed by GNNs are task dependent and supervised, and their application to graphs with thousands of nodes and millions of edges is still hampered by GNN scalability issues that represent an open research question in literature. For this reason, future

works will be aimed at investigating how to efficiently implement deep GNN to obtain deep neural models able to efficiently scale with very big graphs[15,21]. More precisely, even if Elias–Fano-based data structures and the SUSS algorithm proposed in this paper have been designed to efficiently implement RW embedding methods, in future research, we plan to consider their integration in the context of GNNs. Considering the ever-increasing amount of knowledge graphs being constructed in several disciplines, GRAPE may be considered as a powerful, effective, and efficient resource that advances knowledge by performing graph-inference tasks to uncover hidden relationships between concepts or to predict properties and discover structures in complex graphs. However, a limitation of the current implementation of GRAPE is the limited availability of algorithms specifically designed for the analysis of heterogeneous graphs, but we are already working to fill this gap.

GRAPE focuses primarily on CPU models, since most existing GPU Video RAM (VRAM) are too small for several real-world graphs, leading to latency problems as data are moved back and forth between RAM and VRAM. Recently introduced top-tier GPU models provide VRAM that is considerably larger than previously available ones, potentially making it viable to translate the current CPU implementation into a GPU implementation.

Although GRAPE allowed different experimental set-ups to be compared by composing experiments on different graphs, and by using several embedding methods and prediction models, no method systematically outranked other models. To close this knowledge gap, in future work, we plan to run GRAPE with a large-scale grid search to identify task-specific trends for the various combinations of models and their parameters.

## Methods

GRAPE provides a wide spectrum of graph-processing methods, implemented within the Ensmallen module, including node-embedding methods, methods to combine the node embeddings for obtaining edge embeddings, and models for node-label, edge-label, and edge prediction, implemented within the Embiggen module. The graph-processing methods include fast graph loading, multiple graph holdouts, efficient first- and second-order RWs, and triple and corrupted-triple sampling, plus a wide range of graph-processing algorithms that nicely scale with big graphs, using parallel computation and efficient data structures to speed up the computation.

Ensmallen is implemented using Rust, with fully documented Python bindings. Rust is a compiled language gaining importance in the scientific community[17] thanks to its robustness, reliability, and speed. Rust allows threads and data parallelism to be exploited robustly and safely. To further improve efficiency, some core functionalities of the library, such as the generation of pseudo-random numbers and sampling procedures from a discrete distribution, are written in assembly (Supplementary Section 7.2.1 and 7.2.2).

GRAPE currently provides 50 unique node-embedding models (69 considering redundant implementations, important for benchmarks), with 28 being 'from-scratch' implementations and 41 integrated from third-party libraries. The list of available node-embedding methods is constantly growing, with the ultimate goal of providing a complete set of efficient node-embedding models. The input for the various models (for example, RWs and triples) is provided by Ensmallen in a scalable, highly efficient, and parallel way (Fig. 1a). All models were designed according to the "composition over inheritance" paradigm, to ensure a better user experience through increased modularity and polymorphic behavior[45]. More specifically, Embiggen provides interfaces, specific for either the embedding or each of the prediction tasks, that must be implemented by all models. Third-party models, such as PyKeen[10], Karate Club[33], and Scikit-Learn[46] libraries, are already integrated within GRAPE by implementing these interfaces. GRAPE users can straightforwardly create their models and wrap them by implementing the appropriate interface.

GRAPE has a comprehensive test suite. However, to thoroughly test it against many scenarios, we also employed fuzzers, that is, tools that iteratively generate inputs to find corner cases in the library.

In the next section, we describe the succinct data structures used in the library and detail their efficient GRAPE implementation. We then summarize the spectral and matrix factorization; the RW-, the triple-, and the corrupted-triple-based embedding methods and their GRAPE implementation. Then, we describe the edge-embedding methods and the node- and edge-label prediction methods available in GRAPE. Finally, we detail the GRAPE standardized pipelines to evaluate and compare models for graph prediction tasks.

## Succinct data structures for adjacency matrices

In addition to the heavy exploitation of parallelism, the second pillar of our efficient implementation is the careful design of the data structures that uses as little memory as possible and quickly performs operations on them. The naive representation of graphs explicitly stores its adjacency matrix, with a $\mathcal{O}(|V|^2)$ time and memory complexity, $|V|$ being the number of nodes, which leads to intractable memory costs on large graphs. However, since most large graphs are highly sparse, this problem can be mitigated by storing only the existing edges. Often, the adopted data structure is a compressed sparse rows (CSR[47]) matrix, which stores the source and destination indices of existing edges into two sorted vectors. In Ensmallen, we further compressed the graph adjacency matrix by adopting the Elias–Fano succinct data scheme[18], to efficiently store the edges (Supplementary Section 7.1). Since Elias–Fano representation stores a sorted set of integers using memory close to the information-theoretical limit, we defined a bijective map from the graph-edge set and a sorted integer set. To define such encoding, we assigned a numerical identification (ID) from a dense set to each node, and then we defined the encoding of an edge as the concatenation of the binary representations of the numerical IDs of the source and destination nodes. This edge encoding has the appealing property of representing the neighbors of a node as a sequential and sorted set of numeric values and can therefore be employed in the Elias–Fano data structure. Elias–Fano has faster sequential access than random access (Supplementary Section 7.1.1) and is well suited for graph-processing tasks such as retrieving neighbors during RW computation and executing negative sampling using the outbound or inbound node degree scale-free distributions. GRAPE provides both CSR- and Elias–Fano-based data structures for graph representation to allow a time/memory complexity trade-off for processing large graphs.

**Memory complexity.** Elias–Fano is a quasi-succinct data representation scheme, which provides a memory-efficient storage of a monotone list of $n$ sorted integers, bounded by $u$, by using at most $\mathcal{EF}(n, u) = 2n + n\lceil \log_2 \frac{u}{n} \rceil$ bits, which was proven to be less than half a bit per element away from optimality[18] and assures random access to data in average constant time. Thus, when Elias–Fano is paired with the previously presented encoding, the final memory complexity to represent a graph $G(V,E)$ is $\mathcal{EF}_\phi(|V|, |E|) = \mathcal{O}\left(|E| \log \frac{|V|^2}{|E|}\right)$; this is asymptotically better than the $\mathcal{O}(|E| \log |V|^2)$ complexity of the CSR scheme.

**Edge encoding.** Ensmallen converts all the edges of a graph $G(V, E)$ into a sorted list of integers. Considering an edge $e = (v,x) \in E$ connecting nodes $v$ and $x$ represented with, respectively, integers $a$ and $b$, the binary representation of $a$ and $b$ is concatenated through the function $\phi_k(a,b)$ to generate an integer index uniquely representing the edge $e$ itself:

$$\phi_k(a,b) = a\,2^k + b, \text{where } k = \lceil \log_2 |V| \rceil \quad \Rightarrow \quad a = \left\lfloor \frac{\phi_k(a,b) - b}{2^k} \right\rfloor,$$

$$b = \phi_k(a,b) - a\,2^k$$

This implementation is particularly fast because it requires only few bit-wise instructions:

$$\phi_k(a,b) = a << k\,|\,b \quad \Rightarrow \quad a = \phi_k(a,b) >> k,$$

$$b = \phi_k(a,b)\,\&\,(2^k - 1)$$

where $<<$ is the left bit shift, $|$ is the bit-wise OR, and $\&$ is the bit-wise AND (see Supplementary Section 7.1.1 for an example and an implementation of the encoding). Since the encoding uses $2k$ bits, it has the best performances when it fits into a CPU word, which is usually 64 bits on modern computers, meaning that the graph must have less than $2^{32}$ nodes and less than $2^{64}$ edges. However, by using multi-word integers, it can be easily extended to even larger graphs.

**Operations on Elias–Fano.** The aforementioned encoding, when paired with Elias–Fano representation, allows an even more efficient computation of RW samples. Indeed, the Elias–Fano representation allows performing rank and select operations by requiring on average constant time. These two operations were initially introduced by Jacobson to simulate operations on general trees and were subsequently proven fundamental to support operations on data structures encoded through efficient schemes. In particular, given a set of integers $S$, Jacobson defined the rank and select operations as follows[19]:

rank($S, m$) returns the number of elements in $S$ less or equal than $m$

select($S, i$) returns the $i$th smallest value in $S$

As explained below, to speed up computation, we deviate from this definition by defining the rank operation as the number of elements strictly lower than $m$. To compute the neighbors of a node using the rank and select operations, we observe that for every pair of nodes $\alpha,\beta$ with numerical ids $a,b$ respectively, it holds that

$$a\,2^k \le a\,2^k + b < (a+1)\,2^k \quad \Rightarrow \quad \phi_k(a,0) \le \phi_k(a,b) < \phi_k(a+1,0)$$

Thus, the encoding of all the edges with source $\alpha$ will fall in the discrete range

$$[\phi_k(a, 0), \phi_k(a+1, 0)) = [a\,2^k, (a+1)\,2^k)$$

Thanks to our definition of the rank operation and the aforementioned property of the encoding, we can easily derive the computation of the degree $d(a)$ of any node $v$ with numerical ID $a$ for the set of encoded edges $\Gamma$ of a given graph, which is equivalent to the number of outgoing edges from that node:

$$d(a) = \text{rank}(\Gamma, \phi_k(a + 1, 0)) - \text{rank}(\Gamma, \phi_k(a, 0))$$

Moreover, we can retrieve the encoding of all the edges $\Gamma_a$ starting from $v$ encoded as $a$, by selecting every index value $i$ falling within the range $[\phi_k(a,0), \phi_k(a+1,0))$:

$$\Gamma_a = \{\text{select}(\Gamma, i) | \text{rank}(\Gamma, \phi_k(a, 0)) \le i < \text{rank}(\Gamma, \phi_k(a + 1, 0))\}$$

We can then decode the numerical id of the destination nodes from $\Gamma_a$, thus finally obtaining the set of numerical IDs of the neighbors' nodes $N(a)$:

$$N(a) = \{\text{select}(\Gamma, i)\,\&\,(2^k - 1) | \text{rank}(\Gamma, \phi_k(a, 0)) \le i < \text{rank}(\Gamma, \phi_k(a + 1, 0))\}$$

In this way, by exploiting the above integer encoding of the graph and the Elias–Fano data scheme, we can efficiently compute the degree and neighbors of a node using rank and select operations.

**Efficient implementation of Elias–Fano.** The performance and complexity of Elias–Fano heavily rely on the details of its implementation. In this section, our implementation is sketched, to show how

we obtain an average constant time complexity for rank and select operations. A more detailed explanation can be found in Supplementary Section 7.1.

Elias–Fano is essentially aimed at the efficient representation of a sorted list of integers $y_0, …, y_n$ bounded by $u$, that is $\forall\, i \in \{1, …, n-1\}$ it represents $0 \le y_{i-1} \le y_i \le y_{i+1} \le u$.

To this aim, it initially splits each value, $y_i$, into a low-bits part, $l_i$, and a high-bits part, $h_i$, where it can be proven that the optimal split between the high and low bits requires $\left\lfloor \log_2 \frac{u}{n} \right\rfloor$ bits[19].

The lower bits are consecutively stored into a low-bits array $L = [l_1, …, l_n]$, while the high bits are stored in a bit vector $H = [h_1, …, h_n]$, by concatenating the inverted unary encoding, $\mathcal{U}$, of the differences (gaps) between consecutive high-bits parts: $H = [\mathcal{U}(h_1 - 0), \mathcal{U}(h_2 - h_1), …, \mathcal{U}(h_n - h_{n-1})]$. We recall that the inverted unary encoding represents a non-negative integer, $n$, with $n$ zeros followed by a 1; as an example, 5 is represented by 000001 (see Supplementary Figs. 21 and 23 for a more detailed illustration of this scheme).

The rank and select operations on the Elias–Fano representation require two fundamental operations: finding the $i$th 1 or 0 on a bit vector. To perform them in an average constant time, having preset a quantum $q$, we build an index for the zeros, $O_0 = [o_1, …, o_k]$, which stores the position of every $q$ zeros, and an index for the ones, $O_1 = [o_1, …, o_k]$, which similarly stores the position of every $q$ ones.

Thanks to the constructed index, when the $i$th value $v$ must be found, the scan can be started from a position, $o_j$, for $j = \left\lfloor \frac{i}{q} \right\rfloor$, that is already near to the $i$th $v$. Therefore, instead of scanning the whole high-bits array for each search, we only need to scan the high-bits array from position $o_j$ to position $o_{j+1}$.

It can be shown that such scans take an average constant time $\mathcal{O}(q)$ at a low expense of the memory complexity, since we need $\mathcal{O}\left(\frac{n}{q}\log_2 n\right)$ bits for storing the two indexes (Supplementary Section 7.1). Indeed, in our implementation, we chose $q = 1{,}024$, which provides good performance at the cost of a low memory overhead of 3.125% over the high bits and, on average, for every select operation, we need to scan 16 words of memory.

**Available data-structure trade-offs.** GRAPE offers a choice between two data structures, CSR and Elias–Fano, at compile time. The CSR data structure is the default option because of its speed and efficiency in handling common graph operations, such as exploring a node's neighborhood. This structure stores the graph as an array of row pointers, column indices, and non-zero values, providing efficient access to the non-zero elements in sparse adjacency matrices.

On the other hand, Elias–Fano's succinct data structure is primarily effective for representing large graphs because, as mentioned earlier, it requires the least amount of memory without additional assumptions. The Elias–Fano structure is recommended in cases in which the graph size is so big that memory conservation becomes crucial.

While GRAPE provides the option to choose between two data structures, an expert user can add and use any other graph data structure optimized for their specific task.

## Spectral and matrix factorization embedding methods

Spectral and matrix factorization methods start by computing weighted adjacency matrices and may include one or more factorization steps. Next, given a target embedding dimensionality, $k$, these models generally use as embeddings the $k$ eigenvectors or singular vectors corresponding to spectral or singular values of interest.

A description of the spectral and matrix factorization methods implemented in GRAPE is reported in Supplementary Section 8.1.

GRAPE provides efficient parallel methods to compute the initial weighted adjacency matrix of the various implemented methods, which are computed either as dense or sparse matrices depending on how many non-zero values the metrics are expected to generate.

The singular vectors and eigenvectors are currently computed using the state-of-the-art LAPACK library[48], although more scalable methods that compute the vectors using an implicit representation of the weighted matrices are currently under investigation.

## First- and second-order RW-based embedding methods

First- and second-order RW embedding models are shallow neural networks, generally composed of two layers and trained on batches of RW samples. Given a window size, these models learn some properties of the sliding windows on the RWs, such as the co-occurrence of two nodes in each window using Glove[28], the window central node given the other nodes in the window using CBOW[27], or vice versa the nodes in the window from the window central node using SkipGram[27]. The optimal window size value may vary considerably depending on the graph diameter and overall topology. Once the shallow model has been optimized, the weights in either the first or the second layer can be used as node embeddings.

An overview of the RW-based methods implemented in GRAPE is reported in Supplementary Section 8.2.

**Efficient implementation of SkipGram and CBOW models.** GRAPE provides both its own implementations and Keras-based implementations for all shallow neural network models (for example, CBOW, SkipGram, TransE). Nevertheless, since shallow models allow for particularly efficient data-race aware and synchronization-free implementations[32], the from-scratch GRAPE implementations significantly outperform the Keras-based ones, as TensorFlow APIs are too coarse and high level for such fine-grained optimizations. While GPU training is available for the TensorFlow models, their overhead with shallow models tends to be so relevant that from-scratch CPU implementations outperform those based on GPU. Moreover, the embedding of large graphs (such as Wikipedia) does not fit in most GPU hardware memory. Still, Keras-based models allow users to experiment with the open software available in the literature for Keras, including, for example, advanced optimizer and learning rate scheduling methods.

The SkipGram and CBOW models are trained using scale-free negative sampling, which is efficiently implemented using the Elias–Fano data structure rank method.

To obtain reliable embeddings, the training phase of the shallow model would need an exhaustive set of RW samples to be provided for each source node, so as to fully represent the source-node context. When dealing with big graphs, the computation of a proper amount of RW samples needs efficient routines to represent the graph into memory, retrieve and access the neighbors of each node, randomly sample an integer, and, in the case of (Node2Vec) second-order RWs[29], compute the transition probabilities, which must be recomputed at each step of the walk.

The first-order RW is implemented using a SIMD routine for sampling integers (Supplementary Section 7.2.1). When the graph is weighted, another SIMD routine is used to compute the cumulative sum of the unnormalized probability distribution (Supplementary Section 7.2.2). The implementation of the second-order RW requires more sophisticated routines described in the next two sections. After that, we present an approximated weighted and second-order RW that allows dealing with high-degree nodes.

**Implementation of second-order RWs.** Node2Vec is a second-order RW sampling method[29] whose peculiarity relies on the fact that the probability of stepping from one node $v$ to its neighbors considers the preceding step of the walk (Supplementary Figure 27). More precisely, Node2Vec defines the unnormalized transition probability $\pi_{vx}$ of moving from $v$ to any direct neighbor $x$, starting at a previous step from node $t$, as a function of the weight $w_{vx}$ on the edge connecting $v$ and $x(v,x)$, and a search bias $\alpha_{pq}(t,x)$:

$$\pi_{vx} = \alpha_{pq}(t,x)\, w_{vx}$$

The search bias $\alpha_{pq}(t,x)$ is defined as a function of the distance $d(t,x)$ between $t$ and $x$, and two parameters $p$ and $q$, called, respectively, the return and in–out parameters:

$$\alpha_{pq}(t,x) = \begin{cases} \frac{1}{p} & \text{if } d(t,x) = 0 \\ 1 & \text{if } d(t,x) = 1 \\ \frac{1}{q} & \text{if } d(t,x) = 2 \end{cases} \qquad (1)$$

If the return parameter $p$ is small, the walk will be enforced to return to the preceding node; if $p$ is large, the walk will otherwise be encouraged to visit new nodes. The in–out parameter $q$ allows varying smoothly between breadth first search (BFS) and depth first search (DFS) behaviors. Indeed, when $q$ is small, the walk will prefer outward nodes, thus mimicking DFS; it will otherwise prefer inward nodes emulating in this case BFS. Since $\alpha$ must be recomputed at each step of the walk, the algorithm to compute it must be carefully designed to guarantee scalability.

In GRAPE, we sped up its computation by decomposing the search bias $\alpha_{pq}(t,x)$ into the in–out bias $\beta_q(t,x)$, related to the $q$ parameter, and the return bias $\gamma_p(t,x)$, related to $p$:

$$\alpha_{pq}(t,x) = \beta_q(t,x)\gamma_p(t,x) \qquad (2)$$

where the two new biases are defined as

$$\beta_q(t,x) = \begin{cases} 1 & \text{if } d(t,x) \le 1 \\ \frac{1}{q} & \text{if } d(t,x) = 2 \end{cases} \qquad \gamma_p(t,x) = \begin{cases} \frac{1}{p} & \text{if } d(t,x) = 0 \\ 1 & \text{if } d(t,x) > 0 \end{cases} \qquad (3)$$

It is easy to see that equation (2) is equivalent to equation (1).

**Efficient computation of the in–out and return biases.** The in–out bias can be re-formulated to allow an efficient implementation: starting from an edge $(t,v)$, we need to compute $\beta_q(t,x)$ for each $x \in N(v)$, where $N(v)$ is the set of nodes adjacent to $v$ including node $v$ itself.

$$\beta_q(t,x) = \begin{cases} 1 & \text{if } d(t,x) \le 1 \\ \frac{1}{q} & \text{otherwise} \end{cases} \quad \Rightarrow \quad \beta_q(t,x) = \begin{cases} 1 & \text{if } x \in N(t) \\ \frac{1}{q} & \text{otherwise} \end{cases}$$

This formulation (Supplementary Fig. 26) allows us to compute in batch the set of nodes $X_\beta$ affected by the in–out parameter $q$:

$$X_\beta = \left\{ x \,\middle|\, \beta_q(t,x) = \frac{1}{q}, q \ne 1 \right\} = N(v) \setminus N(t)$$

where $N(v)$ are the direct neighbors of node $v$. In this way, the selection of the nodes $X_\beta$ affected by $\beta_q$ simply requires computing the difference of the two sets $N(v) \setminus N(t)$. We efficiently compute $X_\beta$ by using a SIMD algorithm implemented in assembly, leveraging AVX2 instructions that work on node-set representations as sorted vectors of the indices of the nodes (see Supplementary Sections 7.2.1 and 7.2.2 for more details). The return bias $\gamma_p$ can be simplified as

$$\gamma_p(t,x) = \begin{cases} \frac{1}{p} & \text{if } d(t,x) = 0 \\ 1 & \text{otherwise} \end{cases} \quad \Rightarrow \quad \gamma_p(t,x) = \begin{cases} \frac{1}{p} & \text{if } t = x \\ 1 & \text{otherwise} \end{cases}$$

It can be efficiently computed using a binary search for the node $t$ in the sorted vector of neighbors. Summarizing, we re-formulated the transition probability $\pi_{vx}$ of a second-order RW in the following way:

$$\pi_{vx} = \beta_q(t,x)\gamma_p(t,v,x)\, w_{vx}$$

$$\beta_q(t,x) = \begin{cases} 1 & \text{if } x \in N(t) \\ \frac{1}{q} & \text{otherwise} \end{cases} \qquad \gamma_p(t,v,x) = \begin{cases} \frac{1}{p} & \text{if } t = x \\ 1 & \text{otherwise} \end{cases}$$

If $p,q$ are equal to one, the biases can be simplified, so that we can avoid computing them. In general, depending on the values of $p,q$ and on the type of the graph (weighted or unweighted), GRAPE provides eight specialized implementations of the Node2Vec algorithm, to significantly speed up the computation (Supplementary Tables 50 and 51). GRAPE automatically selects and runs the specialized algorithm that corresponds to the choice of the parameters $p,q$ and the graph type. This strategy allows a relevant speed-up. For instance, in the base case ($p = q = 1$ and an unweighted graph), the specialized algorithm runs more than 100 times faster than the most complex one ($p \ne 1$, $q \ne 1$, weighted graph). Moreover, as expected, we observe that the major bottleneck is the computation of the in–out bias (Supplementary Table 51).

**Efficient sampling for Node2Vec RWs.** Sampling from a discrete probability distribution is a fundamental step for computing an RW and can be a notable bottleneck. Many graph libraries implementing the Node2Vec algorithm speed up sampling by using the alias method (see Supplementary Section 7.2.3), which allows sampling in constant time from a discrete probability distribution with the support of cardinality $n$, with a pre-processing phase that scales linearly with $n$.

The use of the alias method for Node2Vec incurs the "memory explosion problem" since the pre-processing phase for a second-order RW on a graph with $|E|$ edges has a support whose cardinality is $\mathcal{O}(\sum_{e_{ij} \in E} \deg(j))$, where $\deg(j)$ is the degree of the destination node of the edge $e_{ij} \in E$.

Therefore, the time and memory complexities needed for pre-processing make the alias method impractical even on relatively small graphs. For instance, on the unfiltered human STRING–PPI graph (19,354 nodes and 5,879,727 edges), it would require 777 GB of RAM.

To avoid this problem, we compute the distributions on the fly. For a given source node $v$, our sampling algorithm applies the following steps:

(1) Computation of the unnormalized transition probabilities to each neighbor of $v$ according to the provided in–out and return biases
(2) Computation of the unnormalized cumulative distribution, which is equivalent to a cumulative sum
(3) Uniform sampling of a random value between 0 and the maximum value in the unnormalized cumulative distribution
(4) Identification of the corresponding index through either a linear scan or a binary search, according to the degree of the node $v$

To compute the cumulative sum efficiently, we implemented an SIMD routine that processes at once in CPU batches of 24 values. Moreover, when the length of the vector is smaller than 128, we apply a linear scan instead of a binary search because it is faster thanks to lower branching and better cache locality. Further details are available in Supplementary Section 7.2.2.

**Approximated RWs.** Because the computational time complexity of the sampling algorithm for either weighted or second-order RWs scales linearly with the degree of the considered source node, computing an exact RW on graphs with high-degree nodes (where 'high' refers to nodes having an outbound degree larger than 10,000) would be impractical, also considering that such nodes have a higher probability to be visited.

To cope with this problem, we designed an approximated RW algorithm, where each step of the walk considers only a sub-sampled set of $k$ neighbors and where the parameter $k$ is set to a value significantly lower than the maximum node degree.

An efficient neighborhood sub-sampling for nodes with degree greater than $k$ requires uniformly sampling unique neighbors whose original order must be maintained. To uniformly sample distinct neighbors in a discrete range [0,$n$], we developed an algorithm (SUSS) that divides the range [0,$n$] into $k$ uniformly spaced buckets and then randomly samples a value in each bucket. The implementation of the algorithm is reported in Supplementary Algorithm 1 (Supplementary Section 7.2.4). After splitting the range [0, …, $n-1$] into $k$ equal segments (buckets) with length $\lfloor$delta/$k\rfloor$, SUSS samples an integer from each bucket by using the Xorshift random number generator. To establish whether the distribution of the integers sampled with SUSS is truly approximating a uniform distribution, we sampled $n = 10,000,000$ integers over [0, …, 10,000], by using both SUSS and by drawing from a uniform distribution in [0, …, 10,000]. We then used the one-sided Wilcoxon signed-rank test to compare the frequencies of the obtained indices, and we obtained a $P$ value of 0.9428, meaning that there is not a statistically significant difference among the two distributions. Therefore, by using a time complexity $\Theta(k)$ and a spatial complexity $\Theta(k)$, SUSS produces reliable approximations of a uniform distribution.

The disadvantage of this sub-sampling approach is that two consecutive neighbors will never be selected in the same sub-sampled neighborhood. Nevertheless, considering that the sub-sampling is repeated at each step of the walk, consecutive neighbors have the same probability of being selected in different sub-samplings.

### Triple-sampling and corrupted-triple-sampling methods

Triple-sampling methods are shallow neural networks trained on triples, ($v,\ell,s$), where {$v,s$} is a node pair composed of a source ($v$) and a destination node ($s$), and $\ell$ is a property of the edge ($v,s$) connecting them. Similar to triple-sampling methods, corrupted-triple-sampling methods are trained on the (true) triples ($v,\ell,s$), but also on corrupted triples, which are obtained by corrupting the original triples by substituting the source and/or destination nodes {$v,s$} with randomly sampled nodes {$v',s'$}, while maintaining the attribute unchanged ($v',\ell,s'$). More details about triple sampling and corrupted-triple-sampling methods are available in Supplementary Sections 8.3 and 8.4.

GRAPE provides a full implementation of first- and second-order LINE triple-sampling methods[31], as well as a Rust parallel implementation of the TransE corrupted-triple-sampling method[32]. Moreover, a large set of corrupted-triple-sampling models is integrated from the PyKeen library. The integrated models include Translating Hyperplanes Embedding (TransH), DistMult, Holographic Embedding (HolE), Automatic Scoring Functions (AutoSF), TransF, TorusE, DistMA, Projection Embedding (ProjE), Convolutional Embedding (ConvE), RESCAL, Quaternion Embedding (QuatE), TransD, ERMLP, CrossE, TuckER, Translating Relationships Embedding (TransR), Paired Relations Embedding (PairRE), Rotate Embedding (RotatE), Complex Embedding (ComplEx), and Box Embedding (BoxE)[10]. We refer to each of the original papers for the extensive explanation. The parameters used for the evaluation of node-embedding models in GRAPE pipelines are available in Supplementary Section 4.1.

### Edge-embedding methods and graph visualization

GRAPE offers an extensive set of methods to compute edge embeddings from node embeddings (for example, concatenation, average, cosine distance, L1, L2, and Hadamard operators[29]), and the choice of the specific edge-embedding operator is left to the user, who can set it through a parameter. To meet the various model requirements, the library provides three implementations of the edge embedding. In the first one, all edge-embedding methods are implemxented as Keras/TensorFlow layers and may be employed in any Keras model. In the second one, all methods are also provided in a NumPy implementation. Finally, the third one uses Rust for models where performance is particularly relevant. For instance, the cosine similarity computation in the Rust implementation is over 250× faster than the analogous NumPy

implementation. Whenever possible, the computation of edge embeddings is executed lazily for a given subset of the edges at a time since the amount of RAM required to explicitly rasterize the edge embedding can be prohibitive on most systems, depending on the edge set cardinality of the considered graph. More specifically, while the lazy generation of edge embeddings is possible during training for only a subset of the supported edge and edge-label prediction models, it is supported for all models during inference.

The library also comes equipped with tools to visualize the computed node and edge embedding and their properties, including edge weights, node degrees, connected components, node types, and edge types. For example, in Fig. 1c, we display the node and edge types of the KGCOVID19 graph and whether sampled edges exist by using the first two components of the t-SNE decomposition of the node/edge embeddings[49].

### Node-label, edge-label, and edge prediction models

GRAPE provides implementations to perform node-label prediction, edge-label prediction, and edge prediction tasks.

All the models devoted to any of the three prediction tasks share the following implementation similarities. Firstly, they all implement the abstract classifier interface and therefore provide straightforward methods for training (fit) and inference (predict and predict_proba).

Secondly, all models are multi-modal; that is, they can receive not only the (user-defined) node- or edge-embedded representation, but also other embeddings computed in multiple ways and therefore carrying different semantics (for example, topological node or edge embeddings or BERT embeddings). For edge prediction and edge-label prediction models, this also generalizes to multiple node-type features, which, if available, are concatenated to the considered node features and to the possibility of computing traditional edge metrics (for example, Jaccard, Adamic–Adar, and so on).

For each task, we make available at least eight models from the literature, adapted to the considered task: five are Scikit-learn-based models, namely random forest, extra trees, decision tree, multi-layer perceptron (MLP), and gradient boosting. The remaining three are TensorFlow-based models, namely GraphSAGE[1], Kipf GCN[50], and a baseline GNN.

As per the node-embedding models, custom and third-party models can be integrated through task-specific Python abstract classes.

Scikit-learn-based models make available all the parameters that are available in the Scikit version. It is straightforward to achieve a substantial speed-up without any modification of the Scikit-learn code by simply using Intel's sklearnex (https://www.intel.com/content/www/us/en/developer/articles/technical/benchmarking-intel-extension-for-scikit-learn.html).

TensorFlow-based models make available parameters to set the number of layers in each provided feature's sub-module and head module.

Visualizations of the Kipf GCN model for node-label, edge-label, and edge prediction tasks are also available (see Supplementary Section 9).

All edge prediction models can be trained by sampling the graph negative edges by following either a uniform or a scale-free distribution; by default, we set a scale-free distribution because it generally produces more informative negative training sets, characterized by a smaller covariate shift with respect to the positive set. This approach still guarantees a negligible number of false-negative edges. The unbalance between positive and negative edges is also a free parameter which may be arbitrarily set: by default, the models are trained using a balanced approach; that is, we sample a number of negative edges equal to the number of positive edges.

In addition to the eight models presented in 'Node-label, edge-label, and edge prediction models', we also make available a multi-modal perceptron model implemented in Rust. This model,

analogously to all other models, supports lazy computation of edge embedding and edge features, but does this in an extensively parallel manner with no additional memory requirement over the model weights. The model optimizer is Nadam. The perceptron is a great baseline for comparison, given its rapid convergence, minimal hardware requirements (no GPUs nor notable RAM requirement), and competitive performance in many considered tasks. Such a model is essential to put into perspective the improvements achieved by more complex and often substantially more expensive models.

Parameters used for the evaluation of edge prediction models in GRAPE pipelines are available in Supplementary Section 4.2.

All of the provided edge-label prediction models support binary and multi-class classification tasks. We currently lack support for multi-label classification tasks, which is being addressed.

All of the provided node-label prediction models support binary, multi-class, and multi-label classification tasks. Parameters used for the evaluation of node-label prediction models in GRAPE pipelines are available in Supplementary Section 4.3.

### Pipelines for the evaluation of graph-prediction tasks

To provide actionable and reliable results, the fair and objective comparative evaluation of datasets, graph embedding, and prediction models is crucial and requires not only specifically designed and real-world benchmark datasets[51] but also pipelines that allow non-expert users to easily test and compare graphs and inference algorithms on the desired graphs.

**FAIR graph retrieval.** GRAPE facilitates FAIR access to an extensive set of graphs and related datasets, including both commonly used benchmark datasets and graphs actively used in biomedical research. Any of the available graphs can be retrieved and loaded with a single line of Python code (Fig. 1b), and their list is constantly expanding, thanks to the generous contributions of GRAPE users. The list of resources currently supported can be found in Supplementary Section 3.1.

Findability and accessibility. Datasets may change locations, versions may appear in more than one location, and file formats may change. Using an ensemble of custom web scrapers, we collect, curate, and normalize the most up-to-date datasets from an extensive resource list (currently over 80,000 graphs). The collected metadata is shipped with each GRAPE release, ensuring that end users can always find and immediately access any available version of the provided datasets.

Interoperability. The graph retrieval phase contains steps that robustly convert data from (even malformed) datasets into general-use tab-separated values (TSV) documents that, while primarily used as graph data, can be used for any desired application case.

Reusability. Once loaded, the graphs can be arbitrarily processed and combined, and used with any of the many embedding and classifier models from either the GRAPE library or any third-party model integrated in GRAPE by implementing the interface described in the following section.

**FAIR evaluation pipelines.** GRAPE provides pipelines for evaluating node-label, edge-label, and edge prediction experiments trained on user-defined embedding features by using task-specific evaluation schemas.

In particular, the evaluation schemas for edge prediction models are K-fold cross-validations and Monte Carlo and connected Monte Carlo (Monte Carlo designed to avoid the introduction of new connected components in the training graph) holdouts. All of the edge prediction evaluation schemas may sample the edges in a uniform or stratified way, with respect to a provided list of edge types. Sampling of negative (non-existing) edges may be executed by following

either a uniform or a scale-free distribution. Furthermore, the edge-prediction evaluation may be performed by using varying unbalance ratios (between existent and non-existent edges) to better gauge the true-negative rate (specificity) and false-positive rate (fallout). Stratified K-fold and stratified Monte Carlo holdouts are also provided for node- and edge-label prediction models.

For all tasks, an exhaustive set of evaluation metrics are computed, including AUROC, AUPRC, balanced accuracy, miss rate, diagnostic odds ratio, markedness, Matthews correlation coefficient, and many others.

All the implemented pipelines have integrated support for differential caching, storing the results of every step of the specific experiment and "smoke tests," that is, for running a lightweight version of the experimental set-up with minimal requirements to ensure execution until completion before running the full experiment.

The pipelines can use any model implementing a standard interface we developed. The interface requires the model to implement methods for training (fit or fit_transform) and inference (predict and predict_proba) plus additional metadata methods (for example, whether to use node types, edge types, and others) which are used to identify experimental flaws and biases. As an example, in an edge-label prediction task using node embeddings, GRAPE will use the provided metadata to check whether the selected node-embedding method also uses edge labels. If so, the node embedding will be recomputed during each holdout. Conversely, if the edge labels are not used in the node-embedding method, it may be computed only once. The choice to recompute the node embedding for each holdout, which may be helpful to gauge how much different random seeds change the performance, is left to the user in this latter case.

To configure one of the comparative pipelines, users have to import the desired pipeline from the GRAPE library and specify the following modular elements:

Graphs: the graphs to evaluate, which can be either graph objects or strings matching the names from graph retrieval.

Graph normalization callback: for some graphs, it is necessary to execute normalization steps and filtering, such as the STRING graphs which can, for example, be filtered at 700 minimum edge weight. For this reason, users can provide this optional normalization callback.

Classifier models: the classifier models to evaluate, which can be either a model implemented in GRAPE or custom models implementing the proper interface.

Node, node type, and edge features: the features to be used to train the provided classifier models. These features can be node-embedding models, implemented in GRAPE, or custom embedding models implementing the node-embedding interface.

Evaluation schema: the evaluation schema to follow for the evaluation.

Given any input graph, each pipeline starts by retrieving it (if the name of the graph was provided) and validating the provided features (checking for NaNs, i.e., Not a Number, constant columns, compatibility with the provided graphs); next, and if requested by the user, it computes all the node embeddings to be used as additional features for the prediction task. Once this preliminary phase is completed, the pipeline starts to iterate and generate holdouts following the provided evaluation schema.

For each holdout, GRAPE then computes the node embeddings required to perform the prediction task (such as, topological node embeddings for a node-label prediction task or topological node embeddings followed by their combination through a user-defined edge-embedding operator to obtain the edge embedding in an edge-prediction task), so that a new instance of the provided classifier models can be fitted and evaluated (by using both the required embedding and, eventually, the additional, label-independent features computed in the preliminary phase). The classifier evaluation is finally performed by computing an exhaustive set of metrics including AUROC, AUPRC,

balanced accuracy, miss rate, diagnostic odds ratio, markedness, Matthews correlation coefficient, and many others.

Interfaces are made available for embedding models, node-label prediction, edge-label prediction, and edge prediction. All models available in GRAPE implement these interfaces, and they can be used as starting points for custom integrations. Many usage examples are available in the library tutorials: https://github.com/AnacletoLAB/grape/tree/main/tutorials.

### Reporting summary
Further information on research design is available in the Nature Portfolio Reporting Summary linked to this article.

## Data availability
GRAPE graph retrieval includes all the graphs used in the Ensmallen benchmarks and the pipeline experiments and are all available from https://github.com/AnacletoLAB/grape. Graphs used for the Ensmallen benchmarks are detailed in Supplementary Section 2. Graphs used for edge and node-label prediction experiments are detailed in Supplementary Section 3. The real-world graphs are downloadable from https://archive.org/download/ctd_20220404/CTD.tar (pre-built CTD), https://archive.org/download/pheknowlator_20220411/PheKnowLator.tar (pre-built biomedical PheKnowLator data) and https://archive.org/download/wikipedia_edge_list.npy/wikipedia_edge_list.npy.gz (pre-built English Wikipedia). More details are available in Supplementary Section 6. The procedures for the construction of the train and test graphs for edge prediction are detailed in Supplementary Section 10.2. Source data are provided with this paper.

## Code availability
All the codes of the experiments presented within the manuscript are publicly available from GitHub repositories. GRAPE can be installed from PyPI: https://pypi.org/project/grape. The source code, reference manual, and tutorials for its usage, alongside several application examples, are available on GitHub: https://github.com/AnacletoLAB/grape. In particular, more than 50 tutorials from which to learn how to use the main functionalities of GRAPE are available from https://github.com/AnacletoLAB/grape/tree/main/tutorials. The code is also available on Zenodo: https://doi.org/10.5281/zenodo.7926104 (ref. 52).

All the scripts to reproduce the experiments showed in the paper are available from GitHub. Ensmallen benchmarks: loading graphs, executing first- and second-order RWs https://github.com/LucaCappelletti94/ensmallen_experiments. Approximated RW experiments: https://github.com/AnacletoLAB/grape/blob/main/tutorials/Comparing%20DeepWalk%20and%20Node2Vec%20running%20on%20exact%20and%20approximated%20random%20walks.ipynb. Experimental comparison of node-embedding methods: (1) edge prediction experiments: https://github.com/AnacletoLAB/grape/blob/main/tutorials/Using%20the%20edge%20prediction%20pipeline.ipynb; (2) node-label prediction experiments: https://github.com/AnacletoLAB/grape/blob/main/tutorials/Using%20the%20node-label%20prediction%20pipeline.ipynb. Comparison of GRAPE with state-of-the-art libraries on big real-world graphs: https://github.com/LucaCappelletti94/embiggen_experiments/tree/master/node2vec_comparisons. The software is delivered under the MIT license.

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

## Acknowledgements

This research was supported by the National Center for Gene Therapy and Drugs based on RNA Technology, PNRR-NextGenerationEU program (G43C22001320007), NIH/National Cancer Institute (U01-CA239108-02), Transition Grant Line 1A Project 'NIMI PARTENARIATI H2020' (PSR2015-1720GVALE_01), the Common Fund, Office of the Director, National Institutes of Health (U01-CA239108-02), the Monarch Initiative, National Institute of Health (1R24OD011883-01), Project PID2021-128970OA-I00 by MCIN/AEI/10.13039/501100011033/ FEDER, and the Director, Office of Science, Office of Basic Energy Sciences of the US Department of Energy under contract no. DE-AC02-05CH11231. The funders had no role in the study design, data collection and analysis, decision to publish, or preparation of the manuscript.

## Author contributions

Conceptualization and methodology: L.C., T.F., G.V., E.C., J.R., and P.N.R.; software (design and implementation): L.C. and T.F. with contributions from V.R., T.C., and J.R.; software (documentation): L.C. and T.F.; data curation and investigation: J.R., P.N.R., V.R., T.C., C.C., and M.J.; supervision: G.V., E.C., P.N.R., and J.R.; funding acquisition: C.M., P.N.R., and G.V.; writing (original draft preparation): G.V., E.C., J.R., and P.N.R.; writing (review and editing): all authors.

## Competing interests

The authors declare no competing interests.

## Additional information

**Correspondence and requests for materials** should be addressed to Giorgio Valentini.

# Reporting Summary

## Statistics

For all statistical analyses, confirm that the following items are present in the figure legend, table legend, main text, or Methods section.

| n/a | Confirmed | |
|---|---|---|
| ☐ | ☒ | The exact sample size ($n$) for each experimental group/condition, given as a discrete number and unit of measurement |
| ☐ | ☒ | A statement on whether measurements were taken from distinct samples or whether the same sample was measured repeatedly |
| ☐ | ☒ | The statistical test(s) used AND whether they are one- or two-sided <br> *Only common tests should be described solely by name; describe more complex techniques in the Methods section.* |
| ☐ | ☒ | A description of all covariates tested |
| ☒ | ☐ | A description of any assumptions or corrections, such as tests of normality and adjustment for multiple comparisons |
| ☐ | ☒ | A full description of the statistical parameters including central tendency (e.g. means) or other basic estimates (e.g. regression coefficient) AND variation (e.g. standard deviation) or associated estimates of uncertainty (e.g. confidence intervals) |
| ☐ | ☒ | For null hypothesis testing, the test statistic (e.g. $F$, $t$, $r$) with confidence intervals, effect sizes, degrees of freedom and $P$ value noted <br> *Give P values as exact values whenever suitable.* |
| ☒ | ☐ | For Bayesian analysis, information on the choice of priors and Markov chain Monte Carlo settings |
| ☒ | ☐ | For hierarchical and complex designs, identification of the appropriate level for tests and full reporting of outcomes |
| ☒ | ☐ | Estimates of effect sizes (e.g. Cohen's $d$, Pearson's $r$), indicating how they were calculated |

*Our web collection on statistics for biologists contains articles on many of the points above.*

## Software and code

Policy information about availability of computer code

| Data collection | All the graphs considered within the manuscript experiments can be downloaded using the GRAPE software available from our GitHub repository (https://github.com/AnacletoLAB/grape). |
|---|---|
| Data analysis | In order to perform the data analysis performed in our work we used our GRAPE open source code and also open source code from other publicly available libraries. <br> In particular we used Python v. 3.7 and Rust v. 1.63.0 and the following open source libraries: <br> networkx v. 2.8.5 <br> igraph v. 0.9.11 <br> csrgraph v. 0.1.28 <br> pecanpy v.2.0.0  https://github.com/LucaCappelletti94/PecanPy <br> node2vec v.0.4.5 https://github.com/LucaCappelletti94/node2vec <br> graphEmbedding v.0.1.0 https://github.com/LucaCappelletti94/GraphEmbedding <br> nodevectors v.0.1.23 https://github.com/LucaCappelletti94/nodevectors <br> snap v.6.0 https://github.com/snap-stanford/snap/tree/0b73cda5f0c9f0dcfd47172eea8be26ba414941a <br> fastnode2vec v. 0.0.6 <br> PyTorch Geometric v.2.2.0 https://github.com/pyg-team/pytorch_geometric <br> PyKeen v.1.10.1 https://github.com/pykeen/pykeen <br> Karateclub v.1.3.4 https://github.com/benedekrozemberczki/karateclub <br> Nodevectors v.0.1.23  https://github.com/VHRanger/nodevectors <br> All the GRAPE code  related to the experiments presented within the manuscript is available from publicly accessible GitHub repositories. <br> Firstly, GRAPE (v. 0.1.30) can be readily installed from PyPI: https://pypi.org/project/grape. |

The source code, reference manual and tutorials for its usage, alongside several application examples, are available on GitHub: https://github.com/AnacletoLAB/grape.

All the scripts to reproduce the experiments showed in the paper are available from GitHub:

a) Ensmallen benchmarks: loading graphs, executing first and second-order random walks: https://github.com/LucaCappelletti94/ensmallen_experiments;

b) Approximated random walk experiments: https://github.com/AnacletoLAB/grape/blob/main/tutorials/Comparing%20DeepWalk%20and%20Node2Vec%20running%20on%20exact%20and%20approximated%20random%20walks.ipynb;

c) Experimental comparison of node embedding methods: (I) Edge prediction experiments: https://github.com/AnacletoLAB/grape/blob/main/tutorials/Using%20the%20edge%20prediction%20pipeline.ipynb, (II) Node-label prediction experiments: https://github.com/AnacletoLAB/grape/blob/main/tutorials/Using%20the%20node-label%20prediction%20pipeline.ipynb;

d) Comparison of GRAPE with state-of-the-art libraries on big real-world graphs: https://github.com/LucaCappelletti94/embiggen_experiments/tree/master/node2vec_comparisons.

The GRAPE software is delivered under the MIT license.

For manuscripts utilizing custom algorithms or software that are central to the research but not yet described in published literature, software must be made available to editors and reviewers. We strongly encourage code deposition in a community repository (e.g. GitHub). See the Nature Portfolio guidelines for submitting code & software for further information.

## Data

Policy information about availability of data

All manuscripts must include a data availability statement. This statement should provide the following information, where applicable:
- Accession codes, unique identifiers, or web links for publicly available datasets
- A description of any restrictions on data availability
- For clinical datasets or third party data, please ensure that the statement adheres to our policy

GRAPE graph retrieval includes all the graphs used in the Ensmallen benchmarks and the pipeline experiments and more than 80,000 graphs are available from https://github.com/AnacletoLAB/grape. Graphs used for the Ensmallen benchmarks are detailed in Supplementary Information Section 2. Graphs used for edge and node-label prediction experiments are detailed in Supplementary Information Section 3. The real world graphs used in Section 2.5 are downloadable from https://archive.org/download/ctd_20220404/CTD.tar (Pre-built CTD), https://archive.org/download/pheknowlator_20220411/PheKnowLator.tar (Pre-built biomedical PheKnowLator data), and https://archive.org/download/wikipedia_edge_list.npy/wikipedia_edge_list.npy.gz (Pre-built English Wikipedia). More details are available in Supplementary Information Section 6. The procedures for the construction of train and test graphs for edge prediction are detailed in Supplementary Information Section 10.2. Source Data for Figures 2, 3, 4 and 5 are available with this manuscript

## Human research participants

Policy information about studies involving human research participants and Sex and Gender in Research.

| | |
|---|---|
| Reporting on sex and gender | NA |
| Population characteristics | NA |
| Recruitment | NA |
| Ethics oversight | NA |

Note that full information on the approval of the study protocol must also be provided in the manuscript.

# Field-specific reporting

Please select the one below that is the best fit for your research. If you are not sure, read the appropriate sections before making your selection.

☒ Life sciences  ☐ Behavioural & social sciences  ☐ Ecological, evolutionary & environmental sciences

For a reference copy of the document with all sections, see nature.com/documents/nr-reporting-summary-flat.pdf

# Life sciences study design

All studies must disclose on these points even when the disclosure is negative.

Sample size | Our resource sw can be applied not only to biological data, but to any data that can be represented by a graph. From this standpoint our porposed sw resource can be applied not only to Life Science studies, Behavorial and social sciences and Ecological and environmental sciences, but to any discipline where the corresponding data can be represented through a graph. In our experiments we used data from different public repositories (including biological data too) and we did not study the sample size since in most cases we analyzed very big graphs, characterized by a high number of nodes and huge number of edges.
We included 44 different data sets for comparing the empirical computational time in classical graph processing tasks across different state-of-the-art graph libraries. The size of each graph varies from a few hundreds to billions of edges in order to evaluate the scalability of software libraries. To our knowledge this is one of the largest esperimental comparison performed to evaluate the performance of graph processing software libraries. We then analyzed in detail the performance of state of the art graph processing libraries on three large real-worlds graphs.

We selected the CTD and Wikipedia graphs since they are two wll-known big graphs largely used by the scientific community. Moreover we used also a biomedical Knowledge Graph generated through the tool PheKnowLator, in order to provide an experimental comparison on a significant and large biomedical graph to show the effectiveness of GRAPE in processing and analyzing big biomedical graphs that are interesting for the biomedcial community.

| Data exclusions | We did not exclude any data from our experiments. We only used the GRAPE sw to remove duplicated data in the analyzed repository. |
|---|---|

**Replication**

GRAPE has been designed to make easily reproducible all the results, also comparing results obtained with different sw resources and libraries, by using the standardized pipelines available in GRAPE. All the machine learning experiments described in the paper can be successfully reproduced using the scripts available from the GRAPE GitHub repository.
In particular experimental comparisons between the different libraries and methods have been repeated from 10 to 30 times, depending on the different computational experiments performed, using multiple hold-out techniques. The results obtained by the GRAPE library are robust and stable as witnessed by the low standard deviation of the measured accuracy on the different test sets generated in the repeated experiments.

**Randomization**

Examples were always randomly assigned to training and test samples, using multiple Montecarlo hold-out techniques in order to obtain statistically robust results. GRAPE offers sw resources to perform these randomization steps in a fully automated way in the context of graph-structured data.

**Blinding**

In our experiments we used publicly available data in graph format where labels associated with nodes (when available) are public. However our experimental procedures to assign samples to groups were completely randomized and in our supervised or semi-supervised prediction tasks always the labels of the test set have been not used for training. Hence from this standpoint our experiments were blind. However, the main aim of our work was not to provide novel experimental results in Life Sciences (or in any other field), but to provide a sw resource that can be robustly applied to the analysis of graphs to obtain reproducible results and that can scale efficiently with big graphs obtaining at the same time prediction results comparable with state of the art methods.

# Reporting for specific materials, systems and methods

We require information from authors about some types of materials, experimental systems and methods used in many studies. Here, indicate whether each material, system or method listed is relevant to your study. If you are not sure if a list item applies to your research, read the appropriate section before selecting a response.

## Materials & experimental systems

| n/a | Involved in the study |
|---|---|
| ☒ | ☐ Antibodies |
| ☒ | ☐ Eukaryotic cell lines |
| ☒ | ☐ Palaeontology and archaeology |
| ☒ | ☐ Animals and other organisms |
| ☒ | ☐ Clinical data |
| ☒ | ☐ Dual use research of concern |

## Methods

| n/a | Involved in the study |
|---|---|
| ☒ | ☐ ChIP-seq |
| ☒ | ☐ Flow cytometry |
| ☒ | ☐ MRI-based neuroimaging |

