## [Peer Review File · Nature Computational Science]

Peer Review Information

Journal: Nature Computational Science

Manuscript Title: GRAPE for Fast and Scalable Graph Processing and random walk-based Embedding

Corresponding author name(s): Giorgio Valentini

Editorial Notes: n/a

Reviewer Comments & Decisions:

Decision Letter, initial version:
--

Dear Professor Valentini,

Thank you for submitting "GraPE: fast and scalable Graph Processing and Embedding" to Nature Computational Science, and we apologize for the delay in reaching a decision on your manuscript. Regretfully, we cannot offer to publish it in its current form.

Among the considerations that arise at this stage are the manuscript's likely interest to a broad range of researchers in computational science, the pressure on space for the various disciplines covered by Nature Computational Science, and the likelihood that a manuscript would seem of great topical interest to those working in the same or related areas of computational science. We do not doubt the technical quality of your work or that it will be of interest to others working in this area of research. However, I regret that we are unable to conclude that the paper provides the sort of substantial practical or conceptual advance that would be of immediate interest to a broad readership of researchers in computational science.

Should future experimental data allow you to address the following point, we would be happy to look at a revised manuscript (unless, of course, something similar has by then been accepted at Nature Computational Science or appeared elsewhere). This includes submission or publication of a portion of this work somewhere else. In the case of eventual publication, the received date would be that of the revised paper.

- Please apply your tool to at least two real-world problems (with comparisons to other tools). This will help show the advantages of your tool and justify the needs to solve the scalability issue for real problems. These real-world problems can be from fields like bioinformatics, physical science, etc.

If you are interested in submitting a suitably revised manuscript in the future or if you have any questions, please contact me.

Thank you for your interest in Nature Computational Science. I am sorry that on this occasion we cannot be more positive.

Best regards,

Jie Pan, Ph.D.
Associate Editor
Nature Computational Science

Author Rebuttal to Initial comments

Dear Dr. Pan,

Please see below our detailed reply to all the points raised by the first reviewer, as well as our reply to the positive comments and suggestions of the second and third reviewer.

Sincerely,

Giorgio Valentini (on behalf of all the authors)

Reviewers' Comments:

Reviewer #1 (Remarks to the Author):

In the manuscript entitled 'GRAPE: Fast and Scalable Graph Graph Processing and Embedding' by Cappelletti et al. the authors present GRAPE, a library for graph processing and representation learning scalable to large graphs. To scale to large graphs, the library extensively relies on parallel computing and efficient data structures. It consists of two core modules: (1) Ensmallen for graph processing operations that heavily relies on Rust, and (2) Embiggen for graph representation and inference methods that implements node embedding methods (Node3Vec, DeepWalk, Glove etc). The library is useful for fast

and scalable implementations of graph loading and node embeddings methods. However, it feels that it heavily relies on other libraries, [...]

Our response: GRAPE is composed of two main modules: Ensmallen and Embiggen. Ensmallen consists of about 2 million lines of completely new code (about 1.5M lines of Python code and about 200K lines of Rust code – results computed with the Tokei tool (<https://docs.rs/tokei/latest/tokei>)), implementing novel efficient data structures and parallel computing techniques to enable scalable graph processing (see Section 4.1 of the manuscript for details). Embiggen not only integrates with existing libraries for knowledge graph embedding methods (e.g. PyKEEN, see Section 4.4 of the manuscript) but also implements from scratch, and in a very efficient way, 26 state-of-the-art embedding methods (e.g. LINE, Node2Vec, DeepWalk, and Glove – see Section 4.2 of the manuscript), which are actively used in the industry and by the scientific community. They were indeed chosen based on their relevance and popularity and, most importantly, for their potential to scale with big graphs, when implemented in an optimised and efficient way.

Thus GRAPE is not based on other libraries, but is designed to allow other libraries to be integrated into the same API to promote fair and accurate benchmarking.

[...] omits the comparison to existing graph learning libraries [...]

Our response: Our work, as detailed below (see answer to issue 1), provides an unprecedented comparison with state-of-the-art graph representation learning libraries and methods, by using a large set of big, real world graphs. This made it possible to show the scalability properties of our novel software resource.

[...] and the choice of algorithms seems groundless, unjustified, and somewhat outdated. The scope seems limited. [...]

Our response: We implemented a large set of algorithms largely used by the scientific community, as witnessed by the popularity of the GitHub site of GRAPE. We particularly focused (as correctly observed by the third reviewer) on random-walk based graph representation learning methods, since in our opinion, these methods are well suited to scale with big graphs (if properly implemented). Indeed, our massive experimental results in Section 2 of the paper clearly support this scalability claim. However, several other popular methods are implemented from scratch or included from other libraries (see Section 4 of the paper).

1. The library is not well motivated and the connections, differences and comparisons to state-of-the art graph learning libraries are missing. In the first place, PyTorch Geometric that implements graph loading, efficient sparse matrix multiplication GNNs, Node2vec etc, but also PyKeen, knowledge graph

embedding libraries (e.g., Pykg2vec), and other frameworks and datasets for evaluation of graph learning algorithms (e.g., OGB, OGB-LSC).

Our response: We performed a massive comparison with state-of-the-art libraries for random-walk based graph representation learning, as demonstrated by the experimental comparison on random-walk generation using 44 (forty-four) different graphs having a number of edges spanning from thousands to about 4 billions (Section 2.3 and Supplementary Section S2). Moreover, we performed experiments for node-label and edge prediction involving 16 different methods using the FAIR pipelines provided by GRAPE for the unbiased comparison of node-label and edge prediction methods (Section 2.4 and 2.5). Finally we performed a comparison of the node2vec GRAPE implementation with state-of-the-art libraries on three big real-world graphs (Wikipedia, CTD and a Knowledge Graph generated with the PheKnowLator framework - Section 2.6).

In sum, to the best of our knowledge it is the largest comparison ever performed on random-walk based graph representation learning libraries and methods.

We did not previously benchmark the Node2Vec implementation available in PyTorch Geometric (PyG) because this implementation does not easily scale with big graphs. As part of our reviewer response, we experimentally found that with Wikipedia, one of the big graphs used in Section 2.6 of our paper, PyG is not able to complete the computation within three days. Moreover on smaller graphs, such as CORA, we experimentally found that the GRAPE implementation of Node2Vec is two orders of magnitude faster than the one available in PyG. This is not surprising since PyG uses code from another library, i.e fastNode2Vec, for the core Node2Vec sampling routines, which we have previously benchmarked and found to be very slow (see Figure 5 of our paper). Moreover PyG's implementation relies on a linear scan of node neighbourhoods to identify whether a node has a neighbour, which can be highly inefficient on large graphs with high-degree nodes. We have now integrated PyG's implementation into GRAPE using the general interface we provide so that others can easily use and compare the two libraries. This integration allows for a FAIR comparison of the two implementations and shows the significant shortcomings of PyG's implementation. However, we can add PyG's results in Section 2.6 of our paper, even if it is evident that PyG is a library with useful features but surely its Node2vec implementation is not well-designed for scaling with big graphs.

About PyKEEN: GRAPE is not provided as an alternative to PyKEEN or other libraries. Indeed, one of the objectives of GRAPE is to integrate other resources in order to provide an environment able to provide a fair comparison between methods and software libraries. In particular we worked with the PyKEEN authors to make the integration seamless. Apart from this, GRAPE also provides many algorithms and utilities that are not available in other libraries, such as advanced graph algebraic operations, sophisticated holdout methods, node centralities, extensive human-readable reports and knowledge graph quality control, and the computation of many graph properties relevant to knowledge modelling choices such as the graph diameter.

2. The name of the method (GRAPE) is unfortunate as there is already a graph-based framework for feature imputation with the same name: You et al. Handling Missing Data with Graph Representation Learning. NeurIPS '20. These frameworks are too closely related to share the name.

Our response: We observe that GRAPE refers to a software resource (and its name was granted precedence on PyPI), while the cited paper refers to a specific algorithm designed to handle missing data with Graph Representation Learning

3. The only knowledge graph embedding (KGE) method implemented is TransE. But for other KGE that are more recent, the library relies on the PyKeen library. It is not clear why other methods such as RotatE, ComplEx, TuckER etc are not implemented but only TransE. Also, there is a Pykg2vec Python library for knowledge graph embedding that is not even cited.

Our response: In our paper we do not claim that GRAPE implements all the existing GRL methods, but we clarified already in the introduction that it focuses on random-walk based GRL methods. This is particularly meaningful since our main goal is to provide scalable methods. Indeed, the software engine of Ensmallen is able to execute massive random-walks in an efficient and scalable way. This allows providing enough random-walk samples for node and edge embeddings, to fuel the novel and efficient GRAPE implementation (written from scratch) of SkipGram, CBOW and Glove algorithms. We also observe there are also several other methods in the GRL literature, not limited to those cited by the reviewer (consider for instance the rich literature on deep Graph Neural Networks), and it seems to us that it is at least curious and strange to demand and claim that a software resource (even as large as GRAPE is) should implement any algorithm for graph processing. Our strategy was: 1) implement in a very efficient way random-walk based methods able to scale with large graphs using limited resources 2) provide a SW environment that can include other SW libraries to provide a FAIR comparison of GRL methods. Probably here the reviewer misunderstood the main aims of our novel GRAPE resource.

4. For node2vec, authors compare to <https://github.com/eliorc/node2vec> among others, but there is also a fast node2vec implementation (<https://github.com/louisabraham/fastnode2vec>) that is not included

Our response: The reviewer seems to have missed the fact that we performed extensive benchmarking. In Section 2.6 we extensively compared GRAPE with fastnode2vec using 3 big graphs, showing that fastnode2vec is significantly slower than GRAPE. In particular, on the Wikipedia graph, the “fast” node2vec implementation of node2vec was not able to complete the computation within 3 days, while GRAPE required about 5 hours. Moreover, we provided extensive experimental results in synthetic and real-world graphs, demonstrating GRAPE's superior performance compared to existing libraries. In particular, we show that GRAPE outperforms FastNode2Vec in terms of running time and memory usage for every graph processing task, ranging from random-walk generation to node embedding.

5. For node-label and edge-label prediction task, the authors run different classifiers (Perceptron, Decision Tree, Random Forest) on top of the node embedding methods. The classifiers are adapted from the Scikit-learn library which is unlikely to scale to large datasets. Additionally, doing node and edge prediction in such a way (node embedding and then running classifier) is unlikely to achieve as good performance as recent advances in graph neural networks and self-supervised learning. The end-to-end approach should be superior to these approaches and such a way of doing node and edge prediction seems outdated.

Our response: The main aim of the GRAPE library is to provide embeddings in an efficient way and not to reimplement all the supervised (or unsupervised) algorithms that can be used on top of the embedded features. However, about the scalability of scikit-learn, we can observe that it is straightforward to achieve a significant speed-up without any modification of the scikit-learn code by simply using Intel's sklearnex: please visit

<https://www.intel.com/content/www/us/en/developer/articles/technical/benchmarking-intel-extension-for-scikit-learn.html#gs.kzwgu0>.

Regarding the reviewer's claim that Graph Neural Networks are "superior" with respect to random-walk based graph representation learning methods, this should be formally evaluated and demonstrated. And this is hard to do, especially with big graphs, since the non-scalability of many GNN is a well-known open issue. Maybe these methods are "outdated", as outlined by the review; however, their smart and efficient implementation can produce meaningful results on big graphs where deep GNN cannot be applied due to their scalability limitations. Moreover, these are methods actively used in the industry and by the scientific community and we believe that one reason for this is their inherent scalability.

Finally, we would like to outline that the perceptron model provided in GRAPE is implemented from scratch in Rust and it is characterised by strong scalability. We would also highlight that, even if this is not the main aim of the GRAPE library, multi-modal GCNs for node-label, edge-label and edge prediction are also available in our library. These multi-modal models, as well as random forests and other machine learning methods can easily integrate node embeddings and features generated by methods available in our library or generated by other libraries (e.g. BERT embeddings of node and node type descriptions)

6. The authors state that one of their goals is "the fair and reproducible comparison of different graph-based methods". However, with this goal in mind OGB and OGB Large-Scale Challenge (LSC) datasets have been created. OGB-LSC is not even cited.

Our response: We are aware of the existence of the OGB-LSC datasets and of course we can cite OGB-LSC, even if only a preprint arxiv version of the paper is available. However we observe that the OGB-LSC challenge has been launched in the second semester of 2022, when we have already performed our experiments for the GRAPE paper and the paper was under submission at Nat Comp Sci. Moreover OGB-

LSC datasets are valuable, but we used equivalent real world data sets such as Wikipedia, CTD and a large, recently developed biomedical knowledge graph to perform our comparative evaluation of the scaling properties of GRAPE (see Section 2.6). We also observe that in the experiments of section 2.3 we used even larger graphs than those used in the OGB-LSC challenge, including also graphs with about 4 billions of edges (Supplementary Table S2).

Finally we observe that OGB-LSC provide datasets for a large scale challenge, while GRAPE is a software resource that provide evaluation pipelines to enable users to compare any library and method using any valuable data set following the FAIR principles of Findability, Accessibility, Interoperability, and Reusability (see Section 2.5 of the paper). These evaluation methods can be applied to any task-specific dataset, allowing users to evaluate the performance of their graph representation learning models, or other methods on their own data, without introducing bias or inconsistency. This is crucial in the field of graph machine learning, which is a rapidly changing and young field where standard datasets may not always be meaningful or representative of the task at hand.

Reviewer #2 (Remarks to the Author):

This paper presents a Rust-based library for efficient and scalable graph processing and embedding library. The authors perform an extensive set of experiments to compare their system with well-known benchmark algorithms. All analyses and comparisons are satisfactory and demonstrate the strength of the proposed system.

A minor note that the length of the paper may exceed the limits and some of the details could be reserved for the SI material.

Our response: We thank the reviewer for their comments. We agree with the reviewer that the paper is too long and should be shortened by removing some details or moving them to the SI material.

Reviewer #3 (Remarks to the Author):

This study presents GRAPE, a software package for graph processing with the focus on random-walk based graph representation learning. GRAPE provides standard benchmark datasets, implementation of various graph representation learning algorithms, and modular pipelines for reproducible research results following FAIR principles. Their implementation demonstrated significant improvements in memory usage, computational efficiency, and scalability, compared with the state-of-the-art competitors. Evaluations are solid and I appreciate detailed description of implemented methods. The authors did

quite substantial work for scalable implementation of random-walk based graph representation learning algorithms.

- 1) *In the method section, however, I wonder if the authors can more highlight their novel contribution for the efficient implementations if there are any. For instance, Elias-Fano data representation is a known technique, but if the use of it for graph representation is novel, we might want to mention it. If not, we may want to cite a relevant work. Especially, I feel we need a citation to Elias-Fano data representation. Similarly, with respect to the implementation of node2vec, it might be good if described sampling and approximation methods are explored in other studies. Or if they haven't considered in other studies, it can be a novel contribution to the community. I believe it would also strengthen the novelty of this study.*

Our response: We thank the reviewer for their comments. We will modify the paper accordingly in order to better highlight the novel contributions for the efficient implementation or random-walk-based embeddings. We will also outline and strengthen the novelty introduced by the approximated random walks and their role in making the embedded representations of nodes and edges even more scalable.

- 2) *For minor comments, I've noticed a couple of editorial mistakes in Introduction (the third paragraph) and section 4.1 (the first paragraph in page 19). If there are other editorial mistakes, please take care of them.*

Our response: We will fix the errors in Introduction and Section 4.1 (thank you) and other typos and mistakes by carefully rereading the paper

Decision Letter, first revision:

Dear Professor Valentini,

Thank you for your correspondence asking us to reconsider our decision on your Resource, "GraPE: fast and scalable Graph Processing and Embedding". After careful consideration we have decided that we would be willing to consider a revised version of your manuscript.

We do appreciate that you are willing to address referee concerns with new experiments, and we are not questioning whether this work would be of interest to researchers working in this field. Your shared revision plan reads reasonable to us. However, given that referee #1, who is an expert on graph representation learning, was negative on the novelty and raised a multitude of concerns on the provided comparisons, we cannot commit to re-review in the absence of a fully revised manuscript that would address all referee concerns in detail.

Along with your revised manuscript, you should also submit a separate point-by-point response to all of the concerns raised by the referees, in each case describing what changes have been made to the

manuscript or, alternatively, if no action has been taken, providing a compelling argument for why that is the case. If we feel that a substantial attempt has been made to address the referees' comments, this response will be sent back to the referees - along with the revised manuscript - so that they can judge whether their concerns have been addressed satisfactorily or otherwise.

Should we receive such a revision, any decision to re-review would depend on the published literature at the time and the extent to which the revisions have addressed the concerns by the reviewers. I should stress, however, that we would be reluctant to trouble our referees again unless we thought that their comments had been addressed in full.

- ensure it complies with our format requirements as set out in our [Guide to Authors](https://www.nature.com/natcomputsci/for-authors).

- state in a cover note the length of the text, methods and figure legends; the number of references and the number of display items.

Please ensure that all correspondence is marked with your Nature Computational Science reference number in the subject line.

Please use the following link to submit your revised manuscript:

[redacted]

We hope to receive your revised paper within four weeks. If you cannot send it within this time, please let us know so that we can close your file. In this event, we will still be happy to reconsider your paper at a later date so long as nothing similar has been accepted for publication at Nature Computational Science or published elsewhere in the meantime. Should you miss the four-week deadline and your paper is eventually published, the received date will be that of the revised, not the original, version.

I look forward to hearing from you soon.

Best regards,

Jie Pan, Ph.D.
Associate Editor
Nature Computational Science

Author Rebuttal, first revision:

Dear Dr. Pan,

According to your suggestions, we provided a fully revised version of the manuscript NATCOMPUTSCI-21-0875B-Z:

Luca Cappelletti, Tommaso Fontana, Elena Casiraghi, Vida Ravanmehr, Tiffany J. Callahan, Carlos Cano, Marcin P. Joachimiak, Christopher J. Mungall, Peter N. Robinson, Justin Reese, and Giorgio Valentini

GRAPE: Fast and Scalable Graph Processing and random walk-based Embedding

As you can see, we slightly modified the title to address one of the concerns of the first reviewer. Indeed we would like to outline already from the title that our work focuses on random-walk based embedding methods, since these methods can better scale on large graphs compared e.g. to deep Graph Neural Networks.

We addressed all the items raised by the reviewers, and we rewrote Abstract, Introduction, several subsections of the Results sections, and the Discussion. We also performed novel experiments, as requested by the first reviewer. We also reduced the length of the Results and Discussion section to observe the length constraints imposed by the journal.

Finally in the Methods Section we also better explained the novel algorithmic contributions and the implementation details of the software resource.

We added a new author, Carlos Cano (Computer Science and Artificial Intelligence Dept. of the University of Granada, Spain), who significantly helped us in this thorough revision. I hope that this is allowed by the policy of Nat. Comp. Sci.

We provided two versions of the manuscript with and without the highlighted (in blue) changes introduced in the revised version with respect to the original.

Please see below our detailed reply to all the items raised by the reviewers.

Sincerely,

Giorgio Valentini (on behalf of all the authors)

Reviewers' Comments:

Reviewer #1 (Remarks to the Author):

In the manuscript entitled 'GRAPE: Fast and Scalable Graph Graph Processing and Embedding' by Cappelletti et al. the authors present GRAPE, a library for graph processing and representation learning scalable to large graphs. To scale to large graphs, the library extensively relies on parallel computing and efficient data structures. It consists of two core modules: (1) Ensmallen for graph processing operations

that heavily relies on Rust, and (2) Embiggen for graph representation and inference methods that implements node embedding methods (Node2Vec, DeepWalk, Glove etc). The library is useful for fast and scalable implementations of graph loading and node embeddings methods. However, it feels that it heavily relies on other libraries, [...]

Our response: In the revised version of the paper, we better clarify that our library is mostly composed of new code that not only allows efficient graph storage, retrieval and processing, but also guarantees an easy integration of other (already existent and efficient) libraries (see Abstract, Introduction, Section 2.1). This promotes a fair experimental comparison between different models and allows any user to test their models against those implemented from scratch in GRAPE.

More precisely, GRAPE is composed of two main modules: Ensmallen and Embiggen. Ensmallen consists of about 1.7 million lines of completely new code (about 1.5M lines of Python code and about 200K lines of Rust code – results computed with the Tokei tool (<https://docs.rs/tokei/latest/tokei/>)), implementing novel efficient data structures and parallel computing techniques to enable scalable graph processing (see Section 4.1 of the manuscript for details). Embiggen not only integrates with existing libraries for knowledge graph embedding methods (e.g. PyKEEN, see Section 4.5 of the manuscript) but also implements from scratch, and in a very efficient way, 26 state-of-the-art embedding methods (e.g. LINE, Node2Vec, DeepWalk, and Glove – see Sections 4.2 and 4.3 of the manuscript), which are actively used in the industry and by the scientific community. These embedding methods were chosen based on their relevance and popularity and, most importantly, to enable them to scale with big graphs, when implemented in an optimized and efficient way.

Thus GRAPE is not based on other libraries, but is designed to allow other libraries to be integrated into the same API to promote fair and accurate benchmarking.

[...] omits the comparison to existing graph learning libraries [...]

Our response: Our work, as detailed below (see answer to issue 1), provides an unprecedented comparison with state-of-the-art graph representation learning libraries and methods (See section 2.2, 2.3, 2.4 and 2.5) by using a large set of big, real world graphs. This made it possible to show the scalability properties of our novel software resource. We also added new experiments with the PyTorch library, as suggested by the reviewer (see below).

[...] and the choice of algorithms seems groundless, unjustified, and somewhat outdated. The scope seems limited. [...]

Our response: We implemented a large set of algorithms (about 30 different embedding algorithms) largely used by the scientific community, as witnessed by the popularity of GRAPE on GitHub. We particularly focused (as correctly observed by the third reviewer) on random-walk based graph representation learning methods, since in our opinion, these methods are well suited to scale with big graphs (if properly implemented). Indeed, our robust experimental results in Section 2 (Results) of the paper clearly support this scalability claim. Taking into account the Reviewer's observation, to better clarify this point we modified the Introduction as well as the title of the paper to "GRAPE: Fast and Scalable Graph Processing and random walk-based Embedding".

However, several other popular methods are implemented from scratch or included from other libraries (see Section 4. Methods of the paper). Moreover, GRAPE also provides many algorithms and utilities that are not available in other libraries, such as advanced graph algebraic operations, sophisticated holdout methods to avoid biased evaluations, node centralities, extensive human-readable reports and knowledge graph quality control, and the computation of many graph properties relevant to knowledge modeling choices such as the graph diameter.

1. The library is not well motivated and the connections, differences and comparisons to state-of-the-art graph learning libraries are missing. In the first place, PyTorch Geometric that implements graph loading, efficient sparse matrix multiplication GNNs, Node2vec etc, but also PyKeen, knowledge graph embedding libraries (e.g., Pykg2vec), and other frameworks and datasets for evaluation of graph learning algorithms (e.g., OGB, OGB-LSC).

Our response: Our library is motivated by 2 main items: 1) Scaling with large graphs through random-walk based embedding methods using limited resources 2) Providing a software (SW) environment that allows the inclusion of other SW libraries to provide a FAIR comparative evaluation of graph representation learning (GRL) methods that are either already implemented/integrated in GRAPE, or that can be easily integrated by implementing the provided interface. In the revised version we modified Abstract, Introduction and Section 2.1 to better highlight these points.

Regarding the second aim, we indeed remark that, being aware of the continuous research efforts devoted to the development of novel, efficient, and effective GRL models, we developed an interface so that any existing GRL library can be easily integrated to make use of the fast graph storage and retrieval provided by the Ensmallen submodule. In this way, GRAPE can be seen as a collector of GRL methods for any user who wants to perform a FAIR comparison. Since our aims might have been not clearly stated in the paper, according to the Reviewer's observation, we better clarify this in the revised version.

Regarding the comparative evaluation, we performed a massive comparison with state-of-the-art libraries for random-walk based graph representation learning, as demonstrated by the experimental comparison on random-walk generation using 44 (forty-four) different graphs having a number of edges spanning from thousands to about 4 billion (Section 2.3 and Supplementary Section S2). Moreover, we performed experiments for node-label and edge prediction involving 16 different methods using the FAIR pipelines provided by GRAPE for the unbiased comparison of node-label and edge prediction methods (Section 2.4). Finally we performed a comparison of the node2vec GRAPE implementation with state-of-the-art libraries on three big real-world graphs (Wikipedia, CTD and a Knowledge Graph generated with the PheKnowLator framework - Section 2.5).

In summary, to the best of our knowledge it is the largest comparison ever performed on random-walk based graph representation learning libraries and methods.

Nevertheless, to further improve the comparison with state-of-the-art graph learning libraries, according to the Reviewer's suggestions, in the revised version of the paper we performed new experiments also with PyTorch Geometric. Results show that the node2vec implementation available in PyTorch Geometric

(PyG) does not easily scale with big graphs. More precisely in the new experiments in Section 2.5 (Scaling with big real-world graphs), PyTorch Geometric went out of memory in the embedding phase, exceeding the available RAM memory (64GB), while GRAPE only requires 54MB with the CTD graph. Moreover on smaller graphs, such as CORA or the STRING PPI graph of *H. sapiens*, we experimentally found that the GRAPE implementation of node2vec is about two orders of magnitude faster than the one available in PyG. This is not surprising since PyG uses code from another library, i.e fastnode2vec, for the core node2vec sampling routines, which we have previously benchmarked and found to be very slow (see Figure 5 of our paper). Moreover PyG's implementation relies on a linear scan of node neighbourhoods to identify whether a node has a neighbour, which can be highly inefficient on large graphs with high-degree nodes. We have now integrated PyG's implementation into GRAPE using the general interface we provide so that others can easily use and compare the two libraries. This integration allows for a FAIR comparison of the two implementations and shows the significant shortcomings of PyG's implementation.

About PyKEEN: GRAPE is not provided as an alternative to PyKEEN or other valuable libraries. On the contrary, one of the objectives of GRAPE is to integrate other resources in order to provide an environment able to provide a fair comparison between methods and software libraries. Indeed we worked with the authors of PyKEEN themselves to obtain a seamless integration between the two libraries.

In the revised version, according to the Reviewer's suggestion, we cite OGB and OGB-LSG as a valuable tool to compare different graph learning algorithms, showing the differences between the OGB and GRAPE evaluation strategies and pipelines (see Introduction and Discussion).

2. The name of the method (GRAPE) is unfortunate as there is already a graph-based framework for feature imputation with the same name: You et al. Handling Missing Data with Graph Representation Learning. NeurIPS '20. These frameworks are too closely related to share the name.

Our response: We observe that GRAPE refers to a software resource (and its name was granted precedence on PyPI), while the cited paper refers to a specific algorithm designed to handle missing data with Graph Representation Learning. Hence it seems to us that it is not mandatory to change the name of the library.

3. The only knowledge graph embedding (KGE) method implemented is TransE. But for other KGE that are more recent, the library relies on the PyKeen library. It is not clear why other methods such as RotatE, ComplEx, TuckER etc are not implemented but only TransE. Also, there is a Pykg2vec Python library for knowledge graph embedding that is not even cited.

Our response: In our paper we do not claim that GRAPE implements all the existing GRL or knowledge graph embedding (KGE) methods. In the revised version of the paper we further clarify that GRAPE focuses on random-walk based GRL methods and as previously stated, we also changed the title to outline this point. This is motivated by the fact that our main goal is to provide scalable methods, while for other less scalable methods (e.g. deep GNN) other valuable SW libraries can be used. Indeed, the software engine of Ensmallen is able to execute massive random-walks in an efficient and scalable way. This allows providing enough random-walk samples for node and edge embeddings, to fuel the novel and efficient GRAPE implementation (written from scratch) of SkipGram, CBOW and GloVe algorithms.

To summarize, our strategy is: 1) to implement in a very efficient way random-walk based methods able to scale with large graphs using limited resources; 2) to provide a SW environment that can include other SW libraries to provide a FAIR comparison of GRL methods.

However the methods cited by the reviewer (*RotatE*, *Complex*, *TuckER*) are integrated into GRAPE from PyKeen using the flexible interface provided by GRAPE itself. Indeed our aim is to easily include valuable resources available from other libraries in a homogeneous and easy-to use SW environment.

4. For *node2vec*, authors compare to <https://github.com/eliorc/node2vec> among others, but there is also a fast *node2vec* implementation (<https://github.com/louisabraham/fastnode2vec>) that is not included

Our response: We performed extensive benchmarking using also the *fastnode2vec* implementation. Indeed in Section 2.5 we extensively compared GRAPE with *fastnode2vec* using 3 big graphs, showing that *fastnode2vec* is significantly slower than GRAPE. In particular, on the Wikipedia graph, the *fastnode2vec* implementation of *node2vec* was not able to complete the computation within 3 days, while GRAPE completed the computation in about 5 hours. Moreover, we provided extensive experimental results in synthetic and real-world graphs, demonstrating GRAPE's superior performance compared to existing libraries. In particular, we show that GRAPE outperforms *fastnode2vec* in terms of running time and memory usage for every graph processing task, ranging from random-walk generation to node embedding.

5. For *node-label* and *edge-label* prediction task, the authors run different classifiers (*Perceptron*, *Decision Tree*, *Random Forest*) on top of the node embedding methods. The classifiers are adapted from the *Scikit-learn* library which is unlikely to scale to large datasets. Additionally, doing node and edge prediction in such a way (node embedding and then running classifier) is unlikely to achieve as good performance as recent advances in graph neural networks and self-supervised learning. The end-to-end approach should be superior to these approaches and such a way of doing node and edge prediction seems outdated.

Our response: The main aim of the GRAPE library is to provide embeddings through an efficient implementation of random walk-based embeddings and not to reimplement all the supervised (or unsupervised) algorithms that can be used on top of the embedded features. However, about the scalability of *scikit-learn*, we can observe that it is straightforward to achieve a significant speed-up without any modification of the *scikit-learn* code by simply using [Intel's *sklearnx*](https://www.intel.com/content/www/us/en/developer/articles/technical/benchmarking-intel-extension-for-scikit-learn.html#gs.kzwgu0): please visit <https://www.intel.com/content/www/us/en/developer/articles/technical/benchmarking-intel-extension-for-scikit-learn.html#gs.kzwgu0>. In the revised version we added the reference to *Intel sklearnx*.

Moreover we would like to outline that the perceptron model provided in GRAPE is implemented from scratch in Rust and it is characterized by strong scalability (from one to two orders of magnitude faster than the perceptron implementation in TensorFlow).

We agree with the reviewer about the advances in graph neural networks and end-to-end approaches, but it is well-known that these methods show relevant scalability issues and cannot be easily applied to large

graphs having billions of edges. This is exactly the reason why we focused on random-walk based GRL methods and their efficient implementation to scale with large graphs.

Moreover, these are methods actively used in industry and by the scientific community and we believe that one reason for this is their inherent scalability.

In the revised version we better clarify this point and the motivation of using random-walk based embedding methods instead of complex deep GNN with large graphs. However the scalable implementation of GNN is cited as an open issue and relevant papers on this topic are cited in the revised version.

We would also highlight that, even if this is not the main aim of the GRAPE library, multimodal GCNs for node-label, edge-label and edge prediction are also available in our library. These multimodal models, as well as random forests and other machine learning methods can easily integrate node embeddings and features generated by methods available in our library or generated by other libraries (e.g. BERT embeddings of node and node type descriptions)

6. The authors state that one of their goals is "the fair and reproducible comparison of different graph-based methods". However, with this goal in mind OGB and OGB Large-Scale Challenge (LSC) datasets have been created. OGB-LSC is not even cited.

Our response: We thank the reviewer for recalling OGB and OGB-LSC. In the revised version we cite OGB-LSC and we outline the valuable characteristics of OGB and the OGB Large Scale Challenge.

The only reason why we did not previously cite the OGB-LSC is that the OGB challenge was launched in the second semester of 2022, when we had already performed our experiments for the GRAPE paper and the paper was under submission at Nat Comp Sci. We also observe that, though OGB-LSC datasets are valuable, we used equivalently valuable real-world data sets to perform our comparative evaluation of the scaling properties of GRAPE (see Section 2.5). Moreover, in the experiments of section 2.3 we used even larger graphs than those used in the OGB-LSC challenge, including also graphs with about 4 billions of edges (Supplementary Table S2).

Finally we observe that OGB-LSC provides datasets specifically designed for a large scale challenge, while GRAPE is a software resource providing evaluation pipelines to enable users to compare any library and method using any valuable data set following the FAIR principles of Findability, Accessibility, Interoperability, and Reusability (see Section 2.5 of the paper). These evaluation methods can be applied to any task-specific dataset, allowing users to evaluate the performance of their graph representation learning models, or other methods on their own data, without introducing bias or inconsistency. In other words, it is our opinion that the two resources are both valuable and can be used in different contexts, having different overall objectives.

Reviewer #2 (Remarks to the Author):

This paper presents a Rust-based library for efficient and scalable graph processing and embedding library. The authors perform an extensive set of experiments to compare their system with well-known benchmark algorithms. All analyses and comparisons are satisfactory and demonstrate the strength of the proposed system.

A minor note that the length of the paper may exceed the limits and some of the details could be reserved for the SI material.

Our response: We thank the reviewer for her/his comments. In the revised version we shortened the paper (e.g. we completely removed the old Section 2.4 and moved most of the Section 2.5.1 to the Supplementary Note S6) and we shortened or removed some redundant paragraphs from the Discussion, and moved some parts from the Results to the Methods section or to the Supplementary Information.

Reviewer #3 (Remarks to the Author):

This study presents GRAPE, a software package for graph processing with the focus on random-walk based graph representation learning. GRAPE provides standard benchmark datasets, implementation of various graph representation learning algorithms, and modular pipelines for reproducible research results following FAIR principles. Their implementation demonstrated significant improvements in memory usage, computational efficiency, and scalability, compared with the state-of-the-art competitors. Evaluations are solid and I appreciate detailed description of implemented methods. The authors did quite substantial work for scalable implementation of random-walk based graph representation learning algorithms.

- 1) In the method section, however, I wonder if the authors can more highlight their novel contribution for the efficient implementations if there are any. For instance, Elias-Fano data representation is a known technique, but if the use of it for graph representation is novel, we might want to mention it. If not, we may want to cite a relevant work. Especially, I feel we need a citation to Elias-Fano data representation. Similarly, with respect to the implementation of node2vec, it might be good if described sampling and approximation methods are explored in other studies. Or if they haven't considered in other studies, it can be a novel contribution to the community. I believe it would also strengthen the novelty of this study.*

Our response: We thank the reviewer for their comments. We accordingly modified the paper in order to better highlight the novel contributions for the efficient implementation of random-walk-based embeddings based on smart data structures. We will also outline and strengthen the novelties introduced by the approximated random walks and their role in making the embedded representations of nodes and edges even more scalable. To this end in Section 4.3.3 we explained more in detail the novel SUSS algorithm that efficiently samples distinct neighbors in a discrete range allowing to significantly speed-up random walks in graphs characterized by very high degree nodes. We also added more citations about Elias-Fano data representation and its application to graph representation.

2) *For minor comments, I've noticed a couple of editorial mistakes in Introduction (the third paragraph) and section 4.1 (the first paragraph in page 19). If there are other editorial mistakes, please take care of them.*

Our response: We fixed the editorial errors in Introduction and Section 4.1 (thank you) and other typos and editorial mistakes by carefully rereading the paper.

Decision Letter, second revision:

Dear Dr. Valentini,

Thank you for submitting your revised manuscript "GraPE: fast and scalable Graph Processing and random walk-based Embedding" (NATCOMPUTSCI-21-0875C). It has now been seen by the original referees and their comments are below. The reviewers find that the paper has improved in revision, and therefore we'll be happy in principle to publish it in Nature Computational Science, pending revisions to satisfy the referees' final requests and to comply with our editorial and formatting guidelines.

TRANSPARENT PEER REVIEW

Nature Computational Science offers a transparent peer review option for original research manuscripts. We encourage increased transparency in peer review by publishing the reviewer comments, author rebuttal letters and editorial decision letters if the authors agree. Such peer review material is made available as a supplementary peer review file. Please state in the cover letter 'I wish to participate in transparent peer review' if you want to opt in, or 'I do not wish to participate in transparent peer review' if you don't. Failure to state your preference will result in delays in accepting your manuscript for publication.

Thank you again for your interest in Nature Computational Science Please do not hesitate to contact me if you have any questions.

Sincerely,

Jie Pan, Ph.D.
Senior Editor
Nature Computational Science

ORCID

Reviewer #1 (Remarks to the Author):

The authors have addressed my concerns.

Reviewer #2 (Remarks to the Author):

Thank you very much for the revised version.

Reviewer #2 (Remarks on code availability):

This project have many contributors and users. I also checked how the issues reported on Github is handled.

Reviewer #3 (Remarks to the Author):

Thanks to the authors for the detail response letter about the significant update in the text.

Although random walk-based graph embeddings predate graph neural network(GNN)-based embeddings, I believe random walk(RW)-based graph embeddings still have practical uses. Maybe, the authors might want to convince readers in this regard. For instance, GNN embeddings are typically task-dependent, while RW-based graph embeddings are task-independent, which can be useful for certain applications (e.g., graph visualization)

Some innovations in this study can be explored for graph neural network (GNN) embeddings. For instance, Elias-Fano encoding of a graph and SUSS graph sampling can be integrated with GNNs. I was wondering if the authors can describe some possibilities in this regard.

For graph representation learning, one of the important graph kernel algorithms can be Weisfeiler-Lehman graph kernel[1], which is provably equivalent to graph convolution neural network. If it makes

sense, the authors might want to include this in the list of state-of-the-art graph representation learning algorithms.

In addition, the evaluation section only mentions CPU processor specifications. I was wondering if GRAPE can run on GPUs as GPUs have become a popular choice to perform machine learning tasks on graphs. If it is beyond the scope of the current work, it might be worth mentioning as a future work.

[1] Shervashidze, Nino, et al. "Weisfeiler-lehman graph kernels." *Journal of Machine Learning Research* 12.9 (2011).

Reviewer #4 (Remarks to the Author):

In this paper, the authors present GRAPE, a software system for random walk-based embedding of graphs aimed at being able to scale to large graphs. GRAPE has 2 main components: Ensmallen, which loads the graphs and executes graph processing operations; Embiggen which implements GRL and inference models. GRAPE also provides interfaces to integrate third-party models and libraries, as well as pipelines to compare and evaluate prediction performances under different experimental settings and utilities for graph visualization.

In the paper, the authors present the results of a very extensive testing of GRAPE. This was done on graphs of different sizes and characteristics, including very large real-world graphs, and they also compare the performance of GRAPE with several state-of-the-art methods.

In my opinion, the results presented clearly demonstrate the value of the system. GRAPE clearly surpasses state-of-the-art methods. It is an extremely useful resource that is currently needed and will greatly benefit the community. In particular, the paper clearly demonstrates the ability of GRAPE to scale to large graphs, in terms of both time and memory usage. The experiments also show how GRAPE allows a proper comparison of graph-based methods and of their software implementations.

Although the paper is easy to read, I felt that it could possibly benefit from reshuffling some sections. Also, overall, the paper is too long (especially the Methods section), and some section could possibly be moved to the Supplementary Material. So, I am detailing below some recommendations for the authors – as I wrote above, the paper is good, and these are just *optional* recommendation that I feel could improve the paper.

1) I feel that some algorithmic novelties are not properly highlighted. In particular:

a. The high performance in loading and in graph processing operations on very large graphs is achieved by exploiting clever data structures and parallelism. The representation is based on the Elias-Fano representation of a sorted set of integers, and it is the first time that I see it used for graph representation. I would provide an intuition for the approach and a short description in the main paper and leave the details to the methods section (these are already there).

b. The high performance of embedding methods is achieved through efficient implementation of random walks and a new algorithm, the Sorted Unique Sub-Sampling. This algorithm, which is introduced in this paper, is particularly useful for processing graphs that contain very high-degree nodes and I note that it is currently described in the main text only in the caption of Figure 3. Again, I would provide an intuition and a brief description in the main paper and leave the details to the methods section (these are already there).

2) The paper includes testing on an impressive number of different datasets. Sometimes I found myself flipping through the different pages of the main paper and the Supp Mat to see their characteristics. It would be nice to include (possibly in the methods section) some sort of table summarizing some of the characteristics of these datasets (I know this is difficult because there are 44+3 datasets!) -- there is currently a table in the Supp Mat. and a description of the 3 large datasets on S6.1

3) I believe that, in figure 3, experiments for panel e were done on the sk-2005 graph, which is different from the graphs used in panels a-d of the same figure. This should be highlighted and motivated/justified (it is a bit confusing).

4) I found this sentence on line 206-207, very cryptic: "We used the Hadamard product for the edge prediction tasks to construct edge embeddings from node embeddings." While I believe I understand what the authors mean, I would expand this.

5) I would also expand lines 233-235 as this will be very useful for the readers – some of it is currently in section 4.8.2 in the Methods section.

6) The Methods sections is very long. In general, I would move large section of the Methods to Supplementary Material. Essentially, I would keep in the Methods sections only parts which are indispensable to understand the algorithms and the datasets. So, for example, I would move to Supp Mat:

a. sections that describe background material (e.g. lines 452-473, or 490-510)

b. sections that introduce GRAPE and are already described elsewhere in the main paper (e.g. lines 337-363).

Few minor points:

- a) line 82: closed parenthesis missing
- b) line 89: should be “uses”
- c) caption Figure 3: third line, should be “performs”
- d) caption Figure 3: fifth line, closed parenthesis missing.
- e) Line 189: should probably be “from scratch” rather than “by”

Reviewer #4 (Remarks on code availability):

I could not review the code yet. But I am happy to do it if you give me a few more days.

Author Rebuttal, second revision:

Reviewer #1 (Remarks to the Author):

The authors have addressed my concerns.

R: We are glad to hear that we have addressed your concerns.

Reviewer #2 (Remarks to the Author): Thank you very much for the revised version. Reviewer #2 (Remarks on code availability):

This project have many contributors and users. I also checked how the issues reported on Github is handled.

R: We thank Reviewer #2 for checking the code availability and issues reported on GitHub. We hope our work will be useful to the scientific community.

Reviewer #3 (Remarks to the Author):

Thanks to the authors for the detail response letter about the significant update in the text.

Although random walk-based graph embeddings predate graph neural network(GNN)-based embeddings, I believe random walk(RW)-based graph embeddings still have practical uses. Maybe, the authors might want to convince readers in this regard. For instance, GNN embeddings are typically task-dependent, while RW-based graph embeddings are task-independent, which can be useful for certain applications (e.g., graph visualization)

R: We completely agree that random walk-based graph embeddings have practical uses and can be useful for certain applications. Random-walk-based methods should be used alongside GNNs and not instead of GNNs. The paper shows that random-walk-based methods are scalable and can efficiently and effectively compute embeddings to be used as input to simple machine learning models. This allows providing scalable solutions to real-world problems that GNNs cannot handle yet. Moreover we also agree with the reviewer that RW-based methods are task independent while GNNs are in most cases task dependent.

We inserted a paragraph in the Discussion section to outline these points.

Some innovations in this study can be explored for graph neural network (GNN) embeddings. For instance, Elias-Fano encoding of a graph and SUSS graph sampling can be integrated with GNNs. I was wondering if the authors can describe some possibilities in this regard.

R: Elias-Fano is a data structure for representing sorted numerical sequences, which can be used to represent data structure for graphs.

It seems to us that its usage in the context of GNNs could be problematic, however we will consider this issue for future work.

About SUSS, in principle it could be applied for graph subsampling also in the context of GNNs, but in our opinion it is likely that there may be better-suited alternatives. However, in the discussion we added this as a new possible research line.

For graph representation learning, one of the important graph kernel algorithms can be Weisfeiler-Lehman graph kernel[1], which is provably equivalent to graph convolution neural network. If it makes sense, the authors might want to include this in the list of state-of-the-art graph representation learning algorithms.

[1] Shervashidze, Nino, et al. "Weisfeiler-lehman graph kernels." *Journal of Machine Learning Research* 12.9 (2011).

R: We agree that the Weisfeiler-Lehman graph kernel is an important graph kernel algorithm, and indeed we already have cited it in the paper (Section 2.4 - page 9: "Experimental comparison of node and edge embedding methods", and in the Supplementary Information too. Moreover we provided its implementation in GRAPE through an integrated version from the Karate Club library.

In addition, the evaluation section only mentions CPU processor specifications. I was wondering if GRAPE can run on GPUs as GPUs have become a popular choice to perform machine learning tasks on graphs. If it is beyond the scope of the current work, it might be worth mentioning as a future work.

R: We agree that GPUs have become popular for graph machine-learning tasks. We explored the use of GPUs for our approach, and our primary goal was to provide a tool that works on large real-world graphs. However, we encountered latency problems with GPUs when the graphs were large and thus could not fit inside the GPU VRAM. Although we tried to minimise the data being moved, the resulting monolithic GPU model we developed, while faster than modular implementations that may be developed using libraries like TensorFlow or PyTorch, was still slower than our CPU model. We are still determining the Pareto boundary between GPUs and CPUs. The more recent GPUs (DGX systems, for instance) with many more GBs of memory and native support for smaller data types (e.g., such as single-byte floats) may resolve this issue entirely. We look forward to testing these models on those high-end devices, and then we leave this for future research work. We added this issue in the Discussion section.

Reviewer #4 (Remarks to the Author):

In this paper, the authors present GRAPE, a software system for random walk-based embedding of graphs aimed at being able to scale to large graphs. GRAPE has 2 main components: Ensmallen, which loads the graphs and executes graph processing operations; Embiggen which implements GRL and inference models. GRAPE also provides interfaces to integrate third-party models and libraries, as well as pipelines to compare and evaluate prediction performances under different experimental settings and utilities for graph visualization.

In the paper, the authors present the results of a very extensive testing of GRAPE. This was done on graphs of different sizes and characteristics, including very large real-world graphs, and they also compare the performance of GRAPE with several state-of-the-art methods.

In my opinion, the results presented clearly demonstrate the value of the system. GRAPE clearly surpasses state-of-the-art methods. It is an extremely useful resource that is currently needed and will greatly benefit the community. In particular, the paper clearly demonstrates the ability of GRAPE to scale to large graphs, in terms of both time and memory usage. The experiments also show how GRAPE allows a proper comparison of graph-based methods and of their software implementations.

Although the paper is easy to read, I felt that it could possibly benefit from reshuffling some sections. Also, overall, the paper is too long (especially the Methods section), and some section could possibly be moved to the Supplementary Material. So, I am detailing below some recommendations for the authors

– as I wrote above, the paper is good, and these are just *optional* recommendation that I feel could improve the paper.

R: We thank the reviewer for his positive comments on our paper. We also appreciate her/his suggestions for improvement and agree that some sections of the paper could be reshuffled and condensed.

1) I feel that some algorithmic novelties are not properly highlighted. In particular:

a. The high performance in loading and in graph processing operations on very large graphs is achieved by exploiting clever data structures and parallelism. The representation is based on the Elias-Fano representation of a sorted set of integers, and it is the first time that I see it used for graph representation. I would provide an intuition for the approach and a short description in the main paper and leave the details to the methods section (these are already there).

b. The high performance of embedding methods is achieved through efficient implementation of random walks and a new algorithm, the Sorted Unique Sub-Sampling. This algorithm, which is introduced in this paper, is particularly useful for processing graphs that contain very high-degree nodes and I note that it is currently described in the main text only in the caption of Figure 3. Again, I would provide an intuition and a brief description in the main paper and leave the details to the methods section (these are already there).

R: In the main paper, we have added more intuitive sentences describing in a few words the innovative algorithms, including the Sorted Unique Sub-Sampling algorithm and we have left the technical details to the Methods section.

2) The paper includes testing on an impressive number of different datasets. Sometimes I found myself flipping through the different pages of the main paper and the Supp Mat to see their characteristics. It would be nice to include (possibly in the methods section) some sort of table summarizing some of the characteristics of these datasets (I know this is difficult because there are 44+3 datasets!) -- there is currently a table in the Supp Mat. and a description of the 3 large datasets on S6.1

R: We agree that a summary table of the dataset characteristics could be helpful. However, due to the large number of datasets, the already provided summary table of 44 graph characteristics may be difficult to fit in the main paper, since the main manuscript is already quite long. Therefore, we have chosen to leave it in the supplementary material.

3) I believe that, in figure 3, experiments for panel e were done on the sk-2005 graph, which is different from the graphs used in panels a-d of the same figure. This should be highlighted and motivated/justified (it is a bit confusing).

R: We modified the text to better motivate the usage of sk-2005 with approximated random walks provided by GRAPE, by highlighting that this graph includes high-degree nodes (degree of several million), whose processing would be very costly or also infeasible with “vanilla” exact random walks.

4) I found this sentence on line 206-207, very cryptic: “We used the Hadamard product for the edge prediction tasks to construct edge embeddings from node embeddings.” While I believe I understand what the authors mean, I would expand this.

R: We have expanded the Hadamard product description in the paper to make it clearer to readers.

5) I would also expand lines 233-235 as this will be very useful for the readers – some of it is currently in section 4.8.2 in the Methods section.

R: Due to shortage of space and word limits, we could not expand the explanation of the standardized pipelines for a FAIR comparison. However, as observed by the reviewer, the standardized pipelines are described in more detail in the Methods section.

6) The Methods section is very long. In general, I would move large section of the Methods to Supplementary Material. Essentially, I would keep in the Methods sections only parts which are indispensable to understand the algorithms and the datasets. So, for example, I would move to Supp Mat:

a. sections that describe background material (e.g. lines 452-473, or 490-510)

b. sections that introduce GRAPE and are already described elsewhere in the main paper (e.g. lines 337-363).

R: We agree with the reviewer. Accordingly, we moved most of Section 4.2 to the Supplementary Information (section S8.1). Then we moved the paragraph “Overview of the implemented methods” to Supplementary Information S8.2. That is, we moved the background information to the Supplementary. Moreover we further shortened the Methods section by moving to the Supplementary Information Section 4.4 Triple-sampling methods and Section 4.5 Corrupted triple-sampling methods, and we wrote

a summary of these two sections in the new Section 4.4. Triple-sampling and corrupted triple sampling methods.

Few minor points:

- a) line 82: closed parenthesis missing
- b) line 89: should be “uses”
- c) caption Figure 3: third line, should be “performs”
- d) caption Figure 3: fifth line, closed parenthesis missing.
- e) Line 189: should probably be “from scratch” rather than “by”

R: We fixed all the minor points, thank you.

Reviewer #4 (Remarks on code availability):

I could not review the code yet. But I am happy to do it if you give me a few more days.

Final Decision Letter:

Dear Professor Valentini,

We are pleased to inform you that your Resource "GRAPE for Fast and Scalable Graph Processing and random walk-based Embedding" has now been accepted for publication in Nature Computational Science.

Once your manuscript is typeset, you will receive an email with a link to choose the appropriate publishing options for your paper and our Author Services team will be in touch regarding any additional information that may be required.

Please note that *Nature Computational Science* is a Transformative Journal (TJ). Authors may publish their research with us through the traditional subscription access route or make their paper immediately open access through payment of an article-processing charge (APC). Authors will not be required to make a final decision about access to their article until it has been accepted. [Find out more about Transformative Journals](https://www.springernature.com/gp/open-research/transformative-journals)

Authors may need to take specific actions to achieve [compliance](https://www.springernature.com/gp/open-research/funding/policy-compliance-faqs) with funder and institutional open access mandates. If your research is supported by a

funder that requires immediate open access (e.g. according to [Plan S principles](https://www.springernature.com/gp/open-research/plan-s-compliance)) then you should select the gold OA route, and we will direct you to the compliant route where possible. For authors selecting the subscription publication route, the journal's standard licensing terms will need to be accepted, including [self-archiving policies](https://www.springernature.com/gp/open-research/policies/journal-policies). Those licensing terms will supersede any other terms that the author or any third party may assert apply to any version of the manuscript.

Acceptance of your manuscript is conditional on all authors' agreement with our publication policies (see <https://www.nature.com/natcomputsci/for-authors>). In particular your manuscript must not be published elsewhere and there must be no announcement of the work to any media outlet until the publication date (the day on which it is uploaded onto our web site).

Before your manuscript is typeset, we will edit the text to ensure it is intelligible to our wide readership and conforms to house style. We look particularly carefully at the titles of all papers to ensure that they are relatively brief and understandable.

Once your manuscript is typeset and you have completed the appropriate grant of rights, you will receive a link to your electronic proof via email with a request to make any corrections within 48 hours. If, when you receive your proof, you cannot meet this deadline, please inform us at rjsproduction@springernature.com immediately.

If you have queries at any point during the production process then please contact the production team at rjsproduction@springernature.com. Once your paper has been scheduled for online publication, the Nature press office will be in touch to confirm the details.

Content is published online weekly on Mondays and Thursdays, and the embargo is set at 16:00 London time (GMT)/11:00 am US Eastern time (EST) on the day of publication. If you need to know the exact publication date or when the news embargo will be lifted, please contact our press office after you have submitted your proof corrections. Now is the time to inform your Public Relations or Press Office about your paper, as they might be interested in promoting its publication. This will allow them time to prepare an accurate and satisfactory press release. Include your manuscript tracking number NATCOMPUTSCI-21-0875D and the name of the journal, which they will need when they contact our office.

About one week before your paper is published online, we shall be distributing a press release to news organizations worldwide, which may include details of your work. We are happy for your institution or funding agency to prepare its own press release, but it must mention the embargo date and Nature Computational Science. Our Press Office will contact you closer to the time of publication, but if you or your Press Office have any inquiries in the meantime, please contact press@nature.com.

We welcome the submission of potential cover material (including a short caption of around 40 words) related to your manuscript; suggestions should be sent to Nature Computational Science as electronic files (the image should be 300 dpi at 210 x 297 mm in either TIFF or JPEG format). We also welcome suggestions for the Hero Image, which appears at the top of our [home page](http://www.nature.com/natcomputsci); these should be 72 dpi at 1400 x 400 pixels in JPEG format. Please note that such pictures should be selected more for their aesthetic appeal than for their scientific content, and that colour images work better than black and white or grayscale images. Please do not try to design a cover with the Nature Computational Science logo etc., and please do not submit composites of images related to your work. I am sure you will understand that we cannot make any promise as to whether any of your suggestions might be selected for the cover of the journal.

Best regards,

Kaitlin McCardle, PhD
Associate Editor
Nature Computational Science